# Spatiotemporal organisation of protein processing in the kidney

Marcello Polesel [1], Monika Kaminska[1], Dominik Haenni[2], Milica Bugarski[1], Claus Schuh[1], Nevena Jankovic[1], Andres Kaech[2], Jose M. Mateos [2], Marine Berquez[3] & Andrew M. Hall [1,4] ✉

The kidney regulates plasma protein levels by eliminating them from the circulation. Proteins filtered by glomeruli are endocytosed and degraded in the proximal tubule and defects in this process result in tubular proteinuria, an important clinical biomarker. However, the spatiotemporal organization of renal protein metabolism in vivo was previously unclear. Here, using functional probes and intravital microscopy, we track the fate of filtered proteins in real time in living mice, and map specialized processing to tubular structures with singular value decomposition analysis and three-dimensional electron microscopy. We reveal that degradation of proteins requires sequential, coordinated activity of distinct tubular sub-segments, each adapted to specific tasks. Moreover, we leverage this approach to pinpoint the nature of endo-lysosomal disorders in disease models, and show that compensatory uptake in later regions of the proximal tubule limits urinary protein loss. This means that measurement of proteinuria likely underestimates severity of endocytotic defects in patients.

The kidney filters small proteins, including peptide hormones, enzymes and carrier proteins, and thus determines their half-life in the circulation[1]. Loss of kidney function substantially alters the plasma proteome[2], leading to adverse consequences[3]. The blood concentration of endogenous proteins like cystatin c is therefore used to estimate kidney function in humans[4]. Meanwhile, impairment of plasma protein reabsorption post filtration results in tubular proteinuria[5] and the appearance in the urine of injury markers like neutrophil gelatinase-associated lipocalin[6], which are used medically as quantitative readouts of PT function and damage[7]. Hence, understanding protein handling in the kidney is critical for correctly interpreting clinical biomarkers.

Filtered proteins bind to two large multi-ligand receptors (megalin and cubilin), and enter PT cells by receptor mediated endocytosis[8]. Receptors and ligands dissociate within acidified endosomes, so the former can be recycled. Endocytosed proteins are further processed within a highly developed and specialized ELS[9], genetic defects in which cause hereditary kidney diseases in humans[5].

However, despite its importance in renal (patho-)physiology, the dynamic workings of the PT ELS in vivo were previously not well delineated, and are not recapitulated by in vitro models[10]. Moreover, the morphology of the ELS changes along the sub-segments (S1-3) of the PT[9], hinting at an axial evolution in function. Early (S1) cells possess a high density of early endosomes (EEs) and contain characteristic large apical vacuoles (LAVs), which have hybrid features of sorting and recycling endosomes[10]. Meanwhile, the high affinity small peptide transporter PEPT2 is expressed more highly in later (S2/3) PT segments[11], implying reliance on upstream processing of the filtrate.

Previous autoradiographic investigations suggested that reabsorbed proteins are rapidly degraded within cathepsin rich lysosomes[12], allowing reclamation of important cargo (e.g., vitamin D[13]) and nutrients (e.g., amino acids[14]). However, this process has not previously been visualized in real time, generating longstanding uncertainty regarding the spatiotemporal kinetics of renal protein metabolism, the precise roles of the different PT segments and the fate

---

[1]Institute of Anatomy, University of Zurich, Zurich, Switzerland. [2]Center for Microscopy and Image Analysis, University of Zurich, Zurich, Switzerland. [3]Institute of Physiology, University of Zurich, Zurich, Switzerland. [4]Department of Nephrology, University Hospital Zurich, Zurich, Switzerland. ✉e-mail: andrew.hall@uzh.ch

of breakdown products[15]. Following intravenous injection of radio-labeled proteins, peptide fragments can be detected in urine[16], but whether these originate from lysosomal catabolism in the PT was unclear[17]. Intravital imaging of fluorescently-labeled proteins enables direct observation of uptake and trafficking in the PT[18], but the close proximity and spatial imbrication of ELS components within cells renders the detection of discrete events challenging. To circumvent this, we have used functional probes and fluorescence unquenching to temporally study protein degradation[19], and applied sophisticated computational analysis to disentangle overlapping processes, and derive their dynamic behavior.

## Results

### Filtered proteins are reabsorbed and degraded in early proximal tubular segments

Intravital multiphoton microscopy was performed in mice injected intravenously with fluorescently-labeled proteins, and S1/S2 were

identified by characteristic autofluorescence signals (Fig. 1a and Supplementary Fig. 1)[20] (S3 cannot be visualized from the cortical surface). Lactoglobulin was chosen as a representative small protein, because it is stable at physiological pH, not biologically active, and contains multiple lysine residues amenable to labeling (Supplementary Fig. 1). To enable high resolution imaging in vivo lactoglobulin was conjugated with Atto dyes that have a high quantum yield and photostability. Injection of free dyes in solution revealed that Atto-532 is not taken up by tubular cells (Supplementary Fig. 1). Characterization in vitro confirmed that enzymatic breakdown of labeled lactoglobulin produced large increases in fluorescence that were dependent on labeling density, but only minimally affected by protein unfolding or solvent pH (Supplementary Fig. 2).

Following intravenous injection, lactoglobulin was filtered rapidly from the blood and reabsorbed in S1 PT segments (Supplementary Fig. 1), as we have previously observed with lysozyme[20]. After a delay of ~10 min, a sustained rise in fluorescence signal

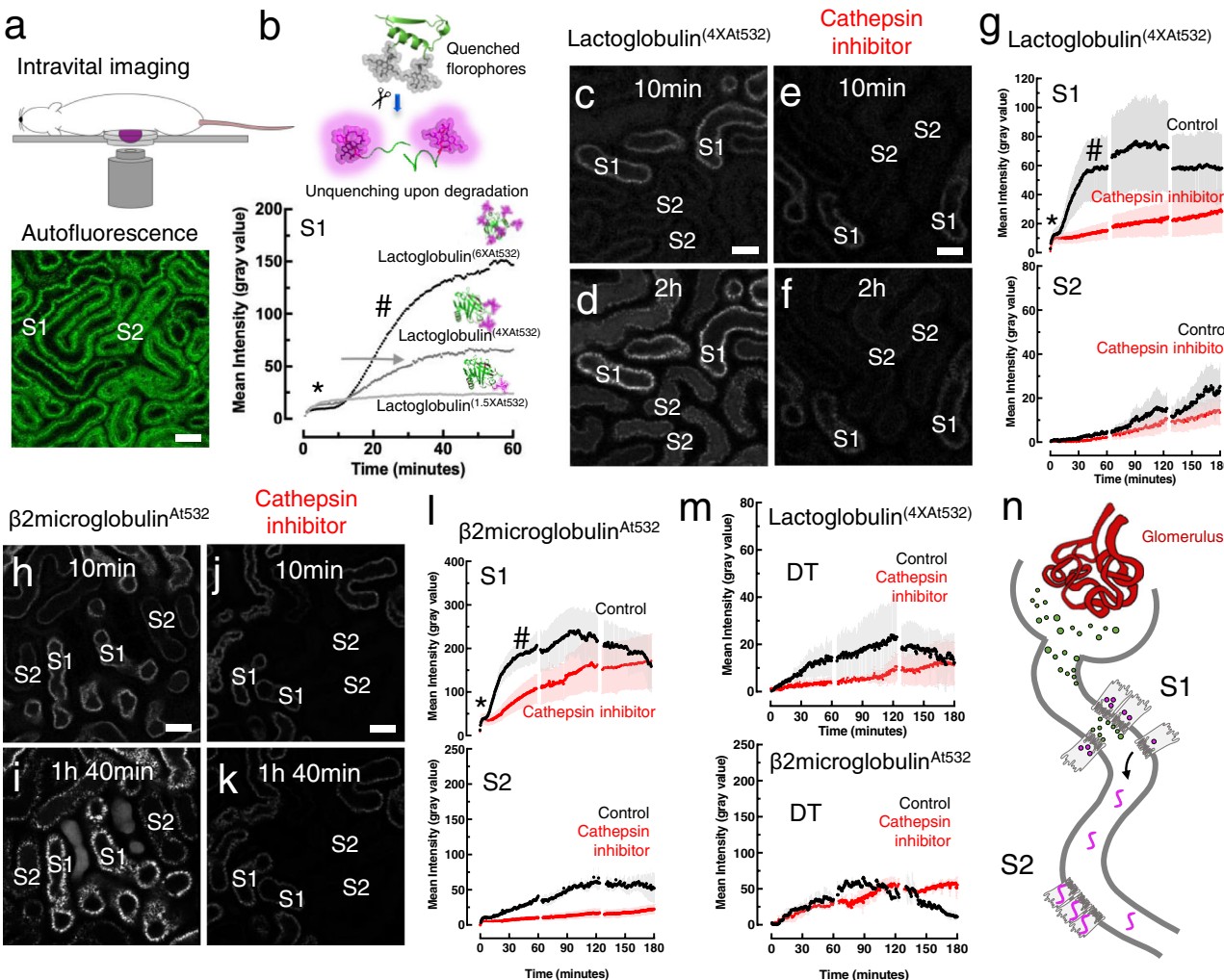

**Fig. 1 | Filtered proteins are endocytosed, degraded and released from S1.**
**a** Early (S1) and late (S2) segments of the proximal tubule (PT) are distinguished in vivo by autofluorescence signals excited at 850 nm (representative image of three independent experiments). **b** Fluorescence unquenching was used to detect protein degradation (#) in S1: signal increase was proportional to labeling density (*=uptake phase). An intermediate labeling density (arrowed) was used for subsequent experiments. Data were derived from regions of interest (ROIs) drawn around whole tubular segments (n = 3 mice). **c**–**l** Uptake and degradation of Lactoglobulin[(4XAt532)] and β2-microglobulin[At532] along the PT. **c**, **h** 10 min after injection, filtration and S1 uptake are complete. **d**, **i** Degradation then commences in S1 and a

delayed fluorescence signal subsequently appeared in S2 segments devoid of initial uptake; these phenomena were severely dampened by a lysosomal cathepsin inhibitor (e64) (**e**, **f** and **j**, **k**). **g**, **l** ROIs were drawn around tubular segments and the plots depict fluorescence intensity over time (mean value ± SEM; n = 3 mice per group). Example single plane images are depicted from the indicated time points (representative of three independent experiments). **m** A delayed signal increase also occurred in the lumen of the distal tubule (DT), with kinetics reflecting S1 degradation (mean value ± SEM; n = 3 mice per group). **n** Summary diagram: filtered proteins are degraded in S1 cells, with release of fragments, some of which undergo a second wave of reabsorption in S2. Scale bars = 20 µm.

occurred, signifying degradation (Fig. 1b–g). The magnitude of the signal increase was proportional to the labeling density (Fig. 1b). To validate findings with a second small protein, we used beta-2-microglobulin and observed similar kinetics (Fig. 1h–l) (a list of the fluorescently-labeled protein tracers used is detailed in Supplementary Table 1). When S1 uptake was inhibited with lysine, lactoglobulin reached S2 and was reabsorbed in this segment (Supplementary Fig. 3), suggesting that it provides a functional reserve.

### Lysosomal degradation products are released and reabsorbed in later tubular segments

Although filtered lactoglobulin did not reach S2 in the absence of lysine, following the onset of unquenching in S1 a delayed appearance of fluorescence signal was observed in S2 cells, and also in the lumens of distal tubules, where the urine is highly concentrated (Fig. 1d, g, i, l, m). The kinetic evolution of these phenomena suggested dependence on lysosomal protein degradation in S1, and supporting evidence for this was provided by a marked dampening effect of pre-treatment with a cathepsin inhibitor (Fig. 1e–g, j–m). Moreover, the rise in S2 signal occurred after most of the lactoglobulin signal had disappeared from the blood.

To further investigate axial handling of protein degradation products along the PT, we enzymatically pre-digested lactoglobulin in vitro (to simulate S1 catabolism), separated the generated fragments according to their size, and fluorescently-labeled them, before injecting intravenously (Fig. 2a). In contrast to whole proteins, injected small peptides (containing 3 or 6 amino acids) passed through S1 and were reabsorbed in S2 (Fig. 2b, d and Supplementary Fig. 4). Co-injection with labeled lysozyme (which is resistant to lysosomal degradation[1]) confirmed that peptide uptake increases dramatically after the point where the vast majority of intact protein has been reabsorbed (Fig. 2f, h, j–m and Supplementary Fig. 4). Conversely, injection of larger peptide fragments (20 amino acids) triggered uptake in S1 (Fig. 2c, e, g, i and Supplementary Fig. 4). Endocytosis of large (20 amino acid) peptide fragments was confirmed in vitro in PT-derived cells (Supplementary Fig. 4).

To assess peptide uptake along the whole PT, kidneys were fixed post intravenous injection, and stained with an established antibody marker of S2 (OAT1) to distinguish segments. This revealed that the highest uptake of small (3 amino acid) peptides was actually in S3 (Supplementary Fig. 4). Moreover, the axial pattern of small peptide uptake was markedly different from that of a 10 kDa labeled dextran (Supplementary Fig. 4), suggesting that the former is not simply reabsorbed by non-specific fluid phase endocytosis.

Next, we cross-referenced these observations to gene expression patterns along the PT, by analyzing the abundance of transcripts relevant to protein metabolism, using an open source database generated from the same strain of mice as our study[21]. This revealed a high expression of endocytotic genes and lysosomal cathepsins in S1 (Fig. 3a), whereas brush border peptidases were much more abundant in S2 (Fig. 3b). Meanwhile, the expression of PEPT2 was low in S1, but increased dramatically in S2, and even more so in S3 (Fig. 3b), matching the uptake pattern of small peptides. Taken together, these findings suggest that peptide fragments generated by lysosomal protein catabolism in S1 are released and undergo further processing in downstream PT segments, which are functionally adapted to this purpose (Fig. 3c).

### Deriving the spatiotemporal kinetics of intracellular protein processing pathways

Having identified S1 as the critical region for initial handling of filtered proteins, we next sought to elucidate in more detail how proteins are processed within the specialized cells of this segment. First, we reconstructed the topography of the S1 ELS in 3-D using focused ion beam/scanning electron microscopy (FIB-SEM) and segmented individual organelles to examine their structure (Fig. 4a–c). This revealed the existence of deep invaginations of the apical brush border membrane, which presumably allow filtered proteins to rapidly penetrate deep within the cell. Lying immediately beneath these invaginations we observed a sub-apical layer of EEs and recycling tubules, with the latter arranged in a network, as described previously in the rat[22]. Finally, a third distinct layer consisting of LAVs and lysosomes was identified approximately in the middle of the cell. Of note, reconstruction of LAVs revealed them to be considerably larger than apparent from 2-D images, and also highly irregular. In contrast, adjacent lysosomes were smaller and displayed a spherical morphology. Antibody staining in fixed sections using ELS markers corroborated the existence of these three overlapping zones (Fig. 4d, e).

To track the progression of proteins through the ELS in vivo we performed line scans across cells following injection of labeled lactoglobulin. To compare data from different cells we normalized the distance from the apical to the basal side from 0 to 1. Due to the spatial overlap of different ELS zones, and the lack of organelle specific molecular markers or an established in vivo model of ELS function, we elected to perform a mathematical approach to disentangle discrete spatial and temporal processes. Singular value decomposition (SVD) analysis is a methodology that deconstructs complex composite signals into individual base vectors[23]. Critically, it does so in an unbiased, model-free manner, and sorts individual components according to their magnitude (singular value). By weighting the spatial and temporal characteristics of each vector with the singular value, their contribution to the overall signal at different time points can then be assessed. Moreover, noisy vectors with low singular values can be removed to generate a cleaner version of the original time series. Thus, SVD analysis is a useful technique for interrogating novel datasets, and has been applied previously to various different imaging modalities, including magnetic resonance imaging[24], computerized tomography scanning[25], photo-acoustic imaging[26], and fluorescence microscopy[27,28].

When applied to our line scan data, SVD analysis revealed the existence of three major, independent base vectors (SVD1, SVD2 and SVD3), which together almost completely recreated the raw kinetic traces (Fig. 5a–d and Supplementary Fig. 5). These findings were remarkably robust across different mice (Supplementary Fig. 5). Moreover, each vector displayed distinct spatial and positional characteristics reminiscent of major ELS structures identified in the preceding structural analysis, namely: LAVs (SVD1), EEs (SVD2) and lysosomes (SVD3) (Fig. 5e, f). However, it is important to note that in the absence of specific molecular markers, these identities cannot be assigned with absolute certainty, and they are probably better conceptualized as distinct, spatially localized processes within the overall pathway of ELS protein processing.

The temporal evolution of the base vectors was then used to shed light on ELS kinetics in vivo. SVD2 displayed an early signal rise and fall in the sub-apical region, likely reflecting rapid transit of proteins through EEs. SVD1 was centered in middle of the cell, in the region of LAVs and lysosomes, but displayed a broad spatial outline more characteristic of the former. The kinetics of SVD1 displayed two distinct phases, an early component starting shortly after protein uptake, and a later component that began after unquenching of proteins. Finally, SVD3 was also centered in the same region as SVD1, but had a much narrower outline, more suggestive of lysosomal morphology. Moreover, signal increase in SVD3 was substantially delayed with respect to SVD1 and 2, and matched temporally with the onset of fluorescence unquenching in whole tubule traces. Taken together, these findings are consistent with older autoradiographic studies showing that proteins transit first through EEs and LAVs before reaching lysosomes[12].

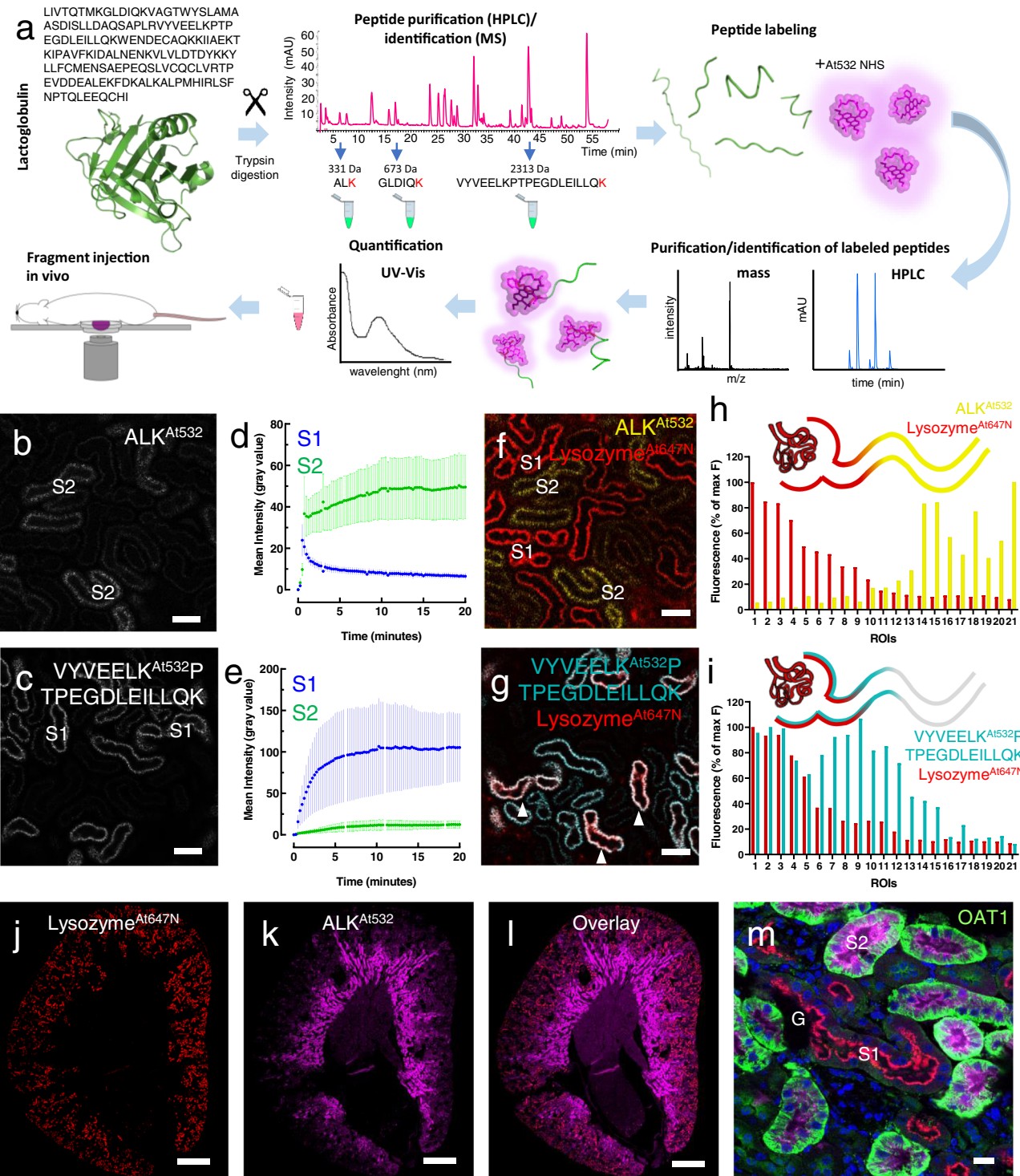

**Fig. 2 | Small peptides bypass S1 and are reabsorbed in S2. a** Enzymatically-digested lactoglobulin peptide fragments of different sizes were labeled with Atto 532 and purified, and their renal handling post intravenous injection was investigated with intravital imaging (mAU milli-absorbance unit). **b** The small peptide ALK^At532 (331 Da) was reabsorbed in S2 (representative of three independent experiments). Scale bar = 20 μm. **c** Conversely, a larger peptide VYVEELK-^At532PTPEGDLEILLQK (2313 Da) triggered S1 uptake (representative of three independent experiments). Scale bars = 20 μm. **d, e** ROIs were drawn around tubular segments and the plots depict fluorescence intensity over time post injection (mean value ± SEM; *n* = 3 mice per group). **f, g** Subsequent injection of intact lysozyme^Atto647N showed a clearly distinct uptake pattern from the small peptide, but not the larger species (representative of three independent experiments;

arrowheads = S1). **h, i** Single experiments showing uptake intensities along the proximal tubule 20 min post injection, with individual ROIs ordered according to lysozyme^Atto647N signal intensity (from highest to lowest). **j–l** Overview of the kidney on cross-section post fixation showing uptake of lysozyme^At647N and ALK^At532 in early and late segments of the proximal tubule, respectively. Scale bars = 500 μm. Single example image planes are depicted (representative of three independent experiments). **m** Higher resolution image of cortical region showing lysozyme^At647N in S1 segments leaving the glomerulus (G), and ALK^At532 in segments staining positive for the S2 marker OAT1. Nuclei were labeled with Hoechst (blue). Scale bar = 20 μm. Single example image planes are depicted (representative of three independent experiments).

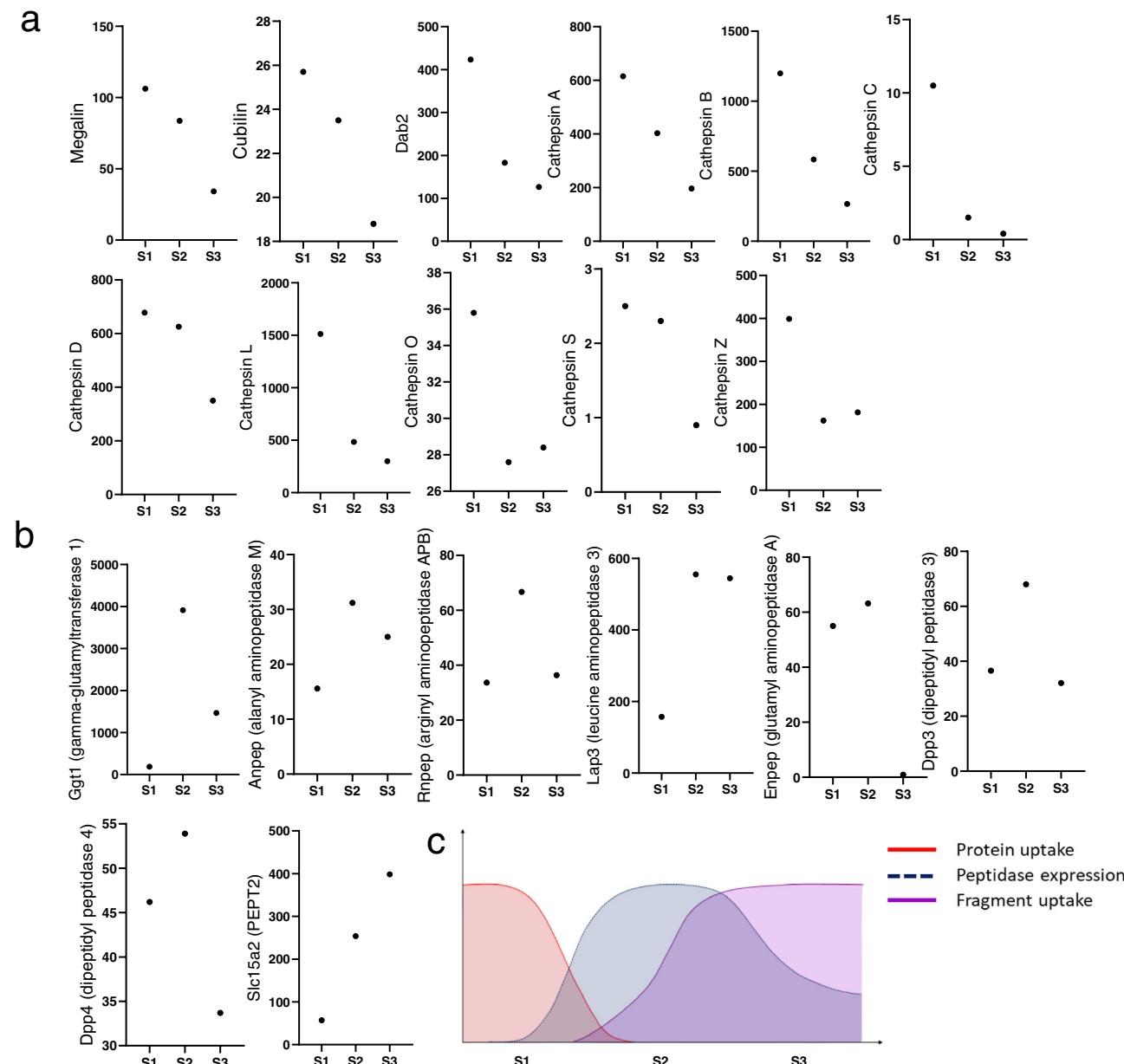

**Fig. 3 | Expression of genes related to protein and peptide handling along the mouse proximal tubule. a** The multi-ligand receptors megalin and cubilin, the endocytotic adaptor protein Dab2, and protein degrading cathepsins are all highly expressed in S1. **b** In contrast, peptidases are more abundant in S2, and the expression of the major renal peptide transporter (PEPT2) increases progressively from S1 to S3. Data depict mRNA levels (TPM values for RNA-Seq) in micro-dissected nephron segments and were derived from ref. 21. **c** Proposed schematic, based on protein/peptide uptake experiments and gene expression data, depicting the spatial arrangement of protein metabolism along the PT.

To provide further validation for this model we performed targeted interventions. Pre-treatment of mice with a cathepsin inhibitor abolished SVD3 (Fig. 5g, h), supporting the notion that this base vector indeed represented lysosomal protein catabolism. However, this did not alter the kinetics of SVD2 or the early component of SVD1, suggesting that these reflect pre-lysosomal trafficking. In contrast, the later component of SVD1 was abolished by cathepsin inhibition. Moreover, this component was attenuated when performing experiments with low-labeled proteins that do not display unquenching upon degradation (Fig. 5i). Conversely, with highly labeled/quenched proteins, the early component of SVD1 was markedly decreased, but the later was accentuated (Fig. 5j). These observations suggest that the first component of SVD1 predominantly reflects pre-lysosomal trafficking of intact proteins into LAVs (i.e., retrograde trafficking from EEs), whereas the

second is dependent on lysosomal protein degradation, and might therefore denote anterograde filling of LAVs from adjacent lysosomes. Since LAVs are linked by an extensive network of recycling tubules to the apical membrane[22], this could provide a potential pathway by which fragments might exit S1 cells. However, we cannot exclude that other processes might contribute, such as lysosomal exocytosis[29].

Lastly, by high resolution time series imaging we observed that the dramatic increase of fluorescence signal post unquenching of highly labeled proteins first occurred in small, punctate structures, before spreading more diffusely within the ELS. By comparing to fixed tissue stained with antibodies to cathepsin L, it could be appreciated that these vesicular structures have a similar morphology to lysosomes (Fig. 5k). Moreover, cross-referencing data to whole tubular signals revealed that the onset of lysosomal protein degradation in S1

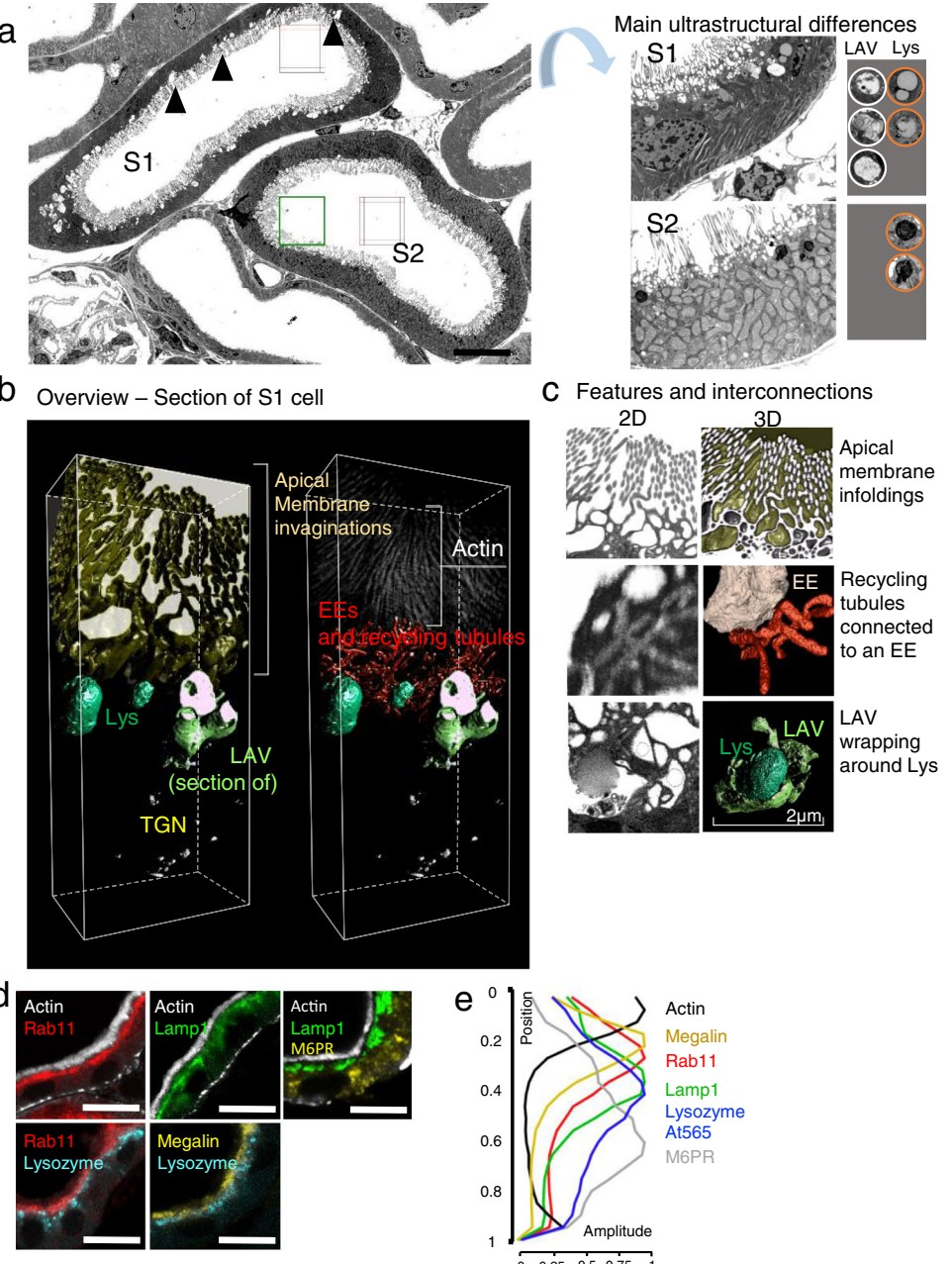

**Fig. 4 | Three-dimensional reconstruction of the S1 endo-lysosomal system.**
**a** 2-D electron microscopy (EM) overview images of S1 and S2 segments of the proximal tubule, from stitched tile sets (boxes), showing large apical vacuoles (LAVs, arrowheads) only in S1. Lysosomes (Lys, orange circles) are more homogeneous than LAVs and appear electron dense in S2. Images acquired from a single animals. Scale bar: 10 μm. **b, c** Reconstruction of the S1 endo-lysosomal system (ELS) in 3-D with FIB-SEM allowed segmentation of different structures (denoted with colors). Deep invaginations of the apical membrane (gold volume) bring the primary urine into close proximity to early endosomes (EEs). Recycling tubules exist in a network between EEs/LAVs and the apical surface. LAVs have a complex, irregular structure, whereas adjacent lysosomes are smaller and more spherical.

Vesicles of the trans-Golgi network (TGN) are observed in the basolateral region of the cell. Images depicted were acquired from a single animal. **d** Fixed kidney tissue sections derived from mice injected with lysozyme[At565] 1 h prior to fixation were stained for established structural markers: actin (apical membrane brush-border), megalin (endocytotic receptor), rab11 (recycling tubules), lamp1 (LAVs/lysosomes), and mannose 6-phosphate receptor (M6PR [TGN]). Single example image planes are depicted (representative of three independent experiments). Scale bars: 10 μm. **e** Line scan analysis revealed distinctive intracellular distributions of the signals, which displayed substantial spatial overlap. The spatial shape is depicted in the position axes (0 = apical side, 1 = basolateral). Amplitudes of markers are from cumulative intensities normalized to 1 ($n \geq 2$ mice).

preceded the appearance of fluorescence in downstream S2 segments, further supporting interdependence of these processes.

In summary, our analysis suggests that protein handling within the PT ELS largely comprises of three major processes (trafficking through EEs, filling of LAVs, and lysosomal degradation), each displaying distinct and reproducible spatial profiles. However, it is important to note that signals from separate structures that share the

same or an inverted kinetic behavior can end up in the same base vectors with SVD analysis. Thus, we cannot ascribe organelle identity with absolute certainty. Nonetheless, we believe that our analysis represents a useful working model of ELS activity in living mice, providing insight into spatiotemporal interplay between organelles, and opening the way to investigate functional changes in disease models in a manner that was not previously possible.

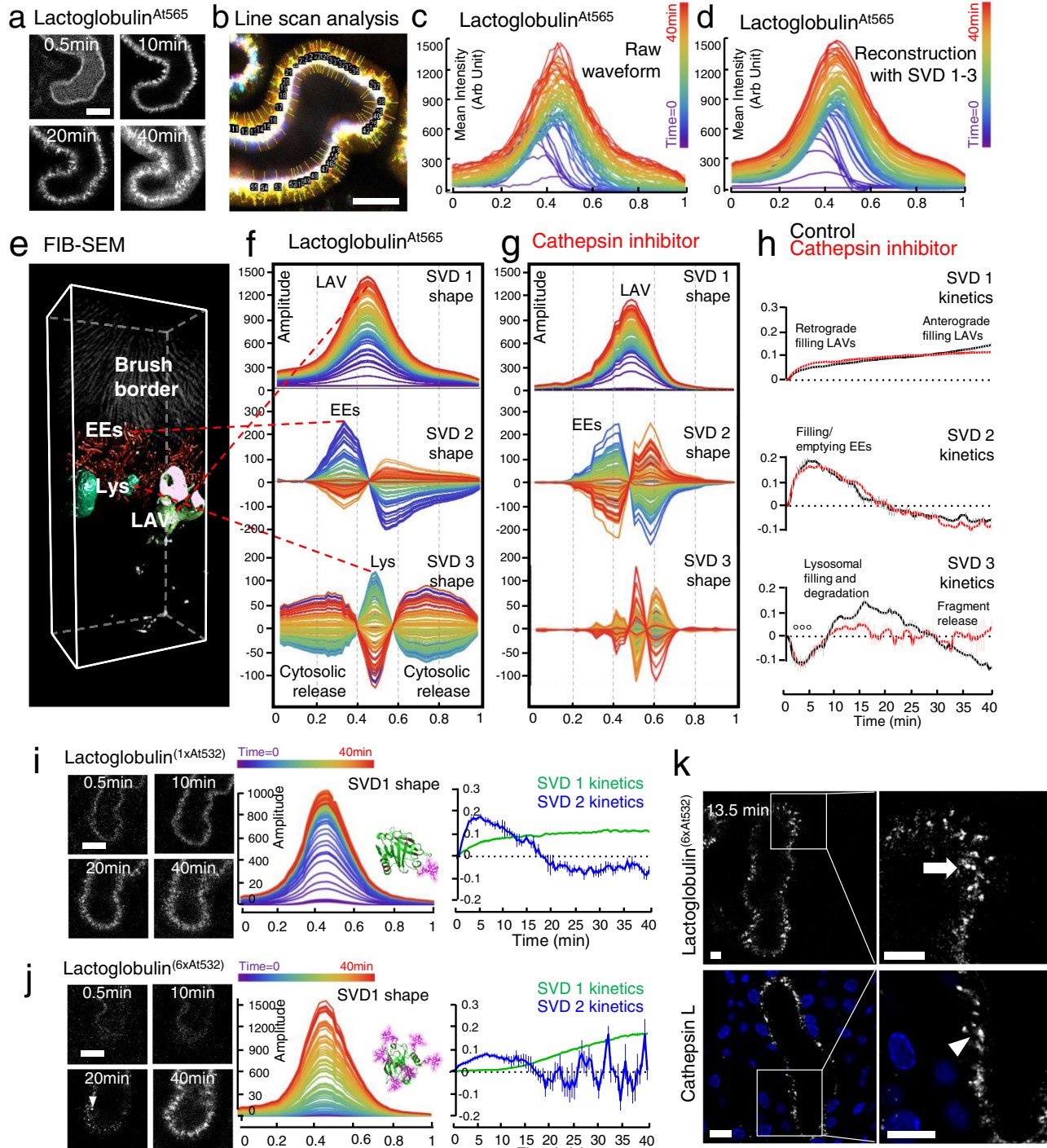

**Fig. 5 | Derivation of protein processing kinetics in S1 cells in vivo. a** High resolution single plane images of S1 cells post intravenous injection of lactoglobulin^At565 (representative of three independent experiments). **b** Line scan ROIs were drawn across cells using sum images (representative of three independent experiments). **c** Representative raw intensity waveform showing normalized evolution of intracellular fluorescence distribution over time (0 = apical side, 1 = basolateral). **d** Reconstruction from three independent base vectors derived from SVD analysis. **e–g** The spatial shape and time evolution of the individual base vectors is depicted (0 = apical side, 1 = basolateral), under control conditions (**f**), and with a cathepsin inhibitor (e64) (**g**). Cross-referencing to the 3-D reconstruction from FIB-SEM suggests that base vectors localize to major endo-lysosomal structures (LAV large apical vacuoles, EE early endosomes, Lys lysosomes). **h** Plots depicting kinetic changes in amplitude of SVD1-3 (mean value ± SEM; *n* = 3 mice per group), and showing loss of lysosomal degradation and fragment release with the cathepsin inhibitor (°°°*p* < 0.001). **i, j** SVD1 displays biphasic kinetics. **i** Post

injection of a low-labeled lactoglobulin^(1xAt532), the first component of SVD1 was clearly visible, but not the second. **j** In contrast, post injection of a high-labeled lactoglobulin^(6xAt532), the second component of SVD1 was accentuated, whereas the amplitude of the first component and SVD2 were markedly attenuated, due the low fluorescence signal emitted from the protein pre-digestion. Following unquenching, a large increase in fluorescence signal was observed in small, punctate vesicles (arrowheads), which subsequently became more diffuse. Single plane images are shown at different time points. Data depicted in plots (mean value ± SEM) are from three experiments. **k** High resolution single plane images (representative of three independent experiments) depicting unquenching of high-labeled lactoglobulin^(6xAt532) in small punctate vesicles (upper panel, arrow), which resemble the structure of lysosomes in proximal tubules in fixed kidney tissue stained for cathepsin L (lower panel, arrowhead). Nuclei were labeled with Hoechst (blue). Scale bars = 10 μm.

### Identification of endo-lysosomal defects and compensatory adaptations in disease models

Several hereditary kidney diseases in humans are caused by genetic defects localizing to the PT ELS[5], with patients developing proteinuria and progressive loss of renal function[30]. Although some of the responsible genes have been identified, detailed understanding of their effects on the endocytotic pathway in vivo was lacking[10]. We therefore applied our approach to two established mouse models. First, we performed experiments in mice lacking *Ocrl* (Lowe oculocerebrorenal syndrome protein, *Ocrl*[y/−])[31], an inositol polyphosphate 5-phosphatase, mutations in which produce Lowe syndrome/Dent 2 disease in humans. We detected a severe impairment (> 50% decrease) in S1 uptake of lactoglobulin (Fig. 6a), consistent with a reported lower expression of megalin[31]. Conversely, SVD analysis showed only subtle abnormalities in protein trafficking and degradation (Fig. 6b–e). In contrast, severe intracellular protein processing defects were observed in mice lacking the Cl⁻/H⁺ exchanger *Clcn5*, the human version of which is mutated in Dent 1 disease[32] (Supplementary Fig. 6). Of note, deletion of *Clcn5* results in substantially more proteinuria than seen in *Ocrl*[y/−][31]. In particular, a striking delay was identified in transit through the apical part of the ELS. Since *Clcn5* is thought to be important for endosomal acidification[33], these phenomena could reflect impaired ligand dissociation from megalin, but further studies will be needed to confirm this.

Compared to healthy animals, the length of PT performing whole protein uptake increased dramatically in *Ocrl*[y/−] mice (Fig. 6f, g). Moreover, axial expression patterns of ELS markers were also markedly altered (Fig. 6h–j). Thus, in the presence of a defect in endocytosis, considerable compensatory reabsorption occurs further along the PT, presumably to limit the loss of proteins and their cargo into the urine (Fig. 6k). Moreover, so long as intracellular protein trafficking and degradation pathways are relatively well preserved, this means that the magnitude of proteinuria massively underestimates the severity of PT uptake disorders, supporting the need for more sensitive urinary biomarkers.

## Discussion

The reabsorption and degradation of filtered plasma proteins in the kidney is a fundamental homeostatic process in living organisms[14], the execution of which necessitates large scale structural and functional adaptations. By tracking events in vivo in time and space (4D imaging) and cross-referencing to ultrastructure and gene expression patterns, we have uncovered an integrated pathway of renal protein processing, involving coordinated activity between different specialized cells and organelles. Our findings suggest that protein metabolism proceeds via a multi-step cascade, and thus contributes to shaping the axial topography of the PT. Spatial separation of tasks within the organ presumably optimizes efficiency; by analogy, protein breakdown in the gut begins in the stomach, but is completed in the intestine.

We introduce a new computational method using SVD analysis to disentangle the kinetics of ELS functions in living animals. As already alluded to, a limitation of this approach is a lack of molecular markers to allow definitive identification of ELS organelles. However, this is a general problem for the field that—along with a paucity of representative in vitro models—has substantially hindered progress to date[10], and provided the motivation to implement a model-free approach. Additional studies using orthogonal techniques will be needed in the future to refine our conclusions, but we note that they are in line with previous estimates from serial EM images[12,34]. Interestingly, our measurements revealed ELS dynamics in the PT to be highly reproducible between mice, perhaps reflecting the environmental selection pressures of a high protein load. Moreover, we demonstrate sufficient sensitivity of the methodology to identify diversification of functional phenotypes caused by single gene deletions, highlighting its usefulness.

A major finding from our study is that intact proteins are catabolized in S1 lysosomes, with subsequent release and reuptake occurring in downstream PT segments. Release of degradation products from S1 cells could explain why peptide fragments can be detected in urine (including in humans), but not when protein uptake in the PT is abolished[35,36]. We also note a previous mass spectrometry imaging study reporting the existence of luminal albumin fragments in mouse kidney sections, which were predicted to arise from cathepsin activity[37]. An alternative explanation of our data could be that delayed signal increase in S2 simply reflects uptake of intact proteins filtered at a much later time point. We consider this unlikely, due to the rapid disappearance of vascular fluorescence signals within 10 min, denoting a fast clearance. However, we cannot exclude the possibility that small amounts of injected proteins remain within the circulation for longer time periods, beyond the limits of detection. Nevertheless, the uptake capacity of S1 is very high and in our experience difficult to saturate[20]. We were therefore only able to demonstrate clear protein uptake in S2 when reabsorption in S1 was strongly inhibited by lysine.

To provide further validation of our model, we used a publicly available gene expression dataset. It is important to acknowledge that gene expression does not always correlate directly with protein abundance, which is a limitation of this readout. Nevertheless, previous histological studies using antibody staining have reported high abundance of lysosomal cathepsins in S1 in rats[38], whereas brush border peptidases are more abundant in S2/3 in mice and rats[39,40], and also in humans[41]. Moreover, the appearance of PT brush border peptidases coincides with that of lysosomal enzymes in post-natal kidney development[42], hinting at an integrated system.

Taken together, our findings suggest that structural segmentation of the PT at least in part reflects adaptation to processing of filtered proteins. While S1 is highly specialized to perform protein reabsorption and degradation, S2 can be conceptualized as a hybrid region, expressing both peptide transporters and endocytotic machinery, and thus capable not only of receiving fragments generated upstream, but also whole proteins when S1 is saturated or dysfunctional. This latter reserve capacity might be particularly important in glomerular diseases when the filtered protein load dramatically increases[43]. Meanwhile, reclaiming released peptides represents a substantial, previously unappreciated function for S3.

By using intravital imaging, we observed evidence of a severe defect in protein endocytosis in *Ocrl*[y/−] mice, which was partially compensated by a substantial increase in uptake length. In contrast, a previous study reported that OCRL depleted PT-derived cell lines proliferate more slowly than control cells, without developing a clear endocytotic phenotype, and that the megalin expressing PT section is shortened in zebrafish lacking ocrl[44]. Further studies will be required to reconcile these discrepancies, but they might be explained by species differences, and also the fact that cell models do not recapitulate the ELS of the PT in vivo. Moreover, it is important to emphasize that the *Ocrl*[y/−] mice are a humanized model, rather than a simple knockout, which complicates the interpretation of the direct effect of *Ocrl* on ELS function. However, a very recent study in mice lacking another ELS gene (*Ehd1*) also reported increased protein uptake length in the PT[45], perhaps suggesting that this is a generic adaptive response, as predicted by modeling of endocytotic defects in humans[44].

There are some further potential limitations to our approach. Labeling of proteins can in theory alter chemical properties, but results were consistent with various different combinations of ligands and tags. Nevertheless, we cannot be certain that findings can be extrapolated to all plasma proteins, and albumin might be handled differently due to binding to cubilin or the neonatal Fc receptor[46,47]. We also cannot exclude that some undetected fragments may be directly reabsorbed into the blood from S1 cells. To achieve sufficient signal intensity, we injected supra-physiological amounts of protein; however, subsequent rapid uptake and metabolism in S1 simply

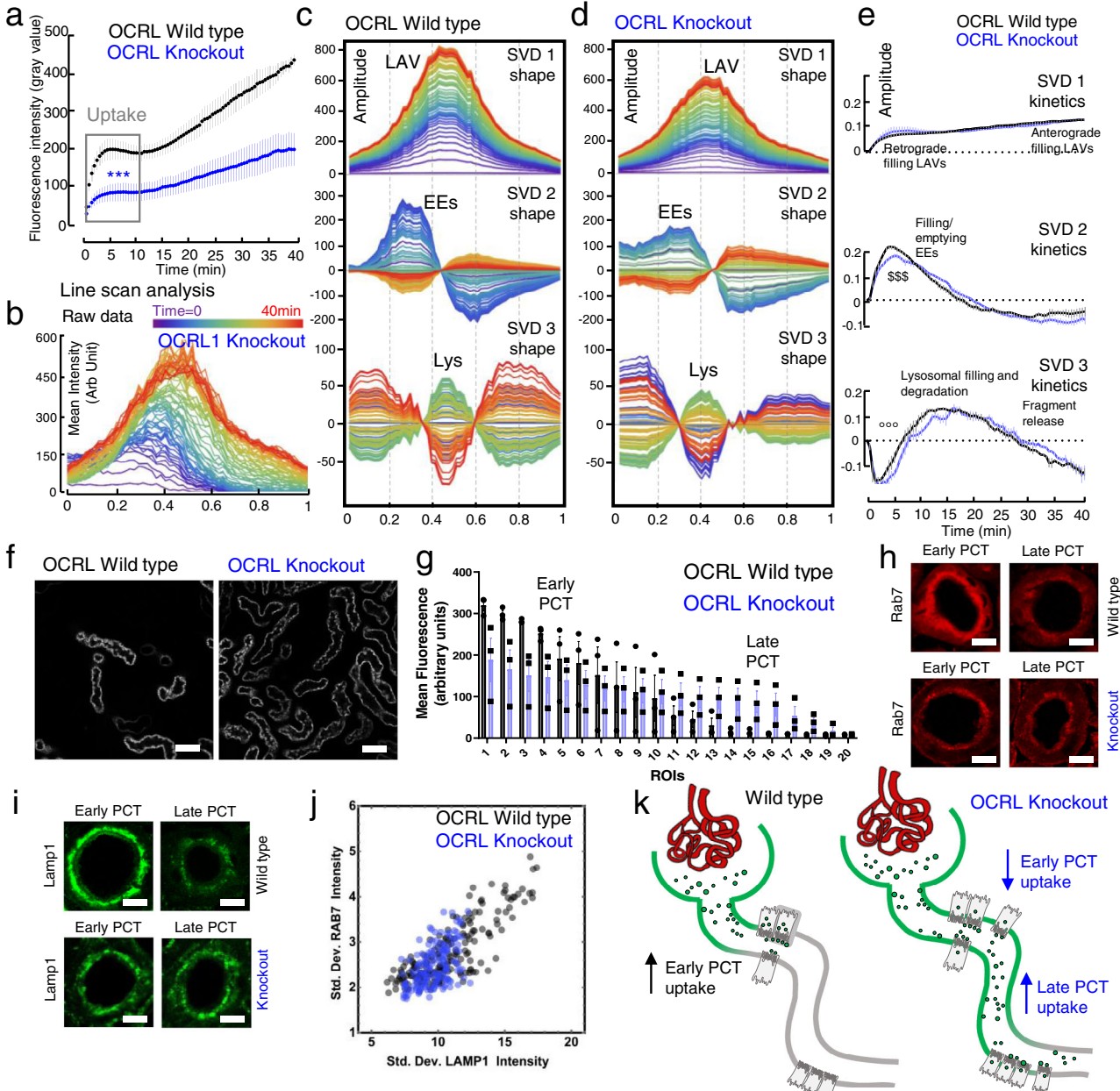

**Fig. 6 | Alterations in endo-lysosomal function and compensatory remodeling in disease. a** Plots depict fluorescence intensity in S1 segments post injection of lactoglobulin$^{At565}$. Uptake was decreased in OCRL knockout (KO) mice compared to wild type (WT) (mean value ± SEM; time × genotype ***$p < 0.001$; $n = 3$ mice per group). **b** Normalized raw intensity waveform derived from line scan ROIs showing intracellular evolution of fluorescence in KO mice (0 = apical, 1 = basolateral). Base vectors derived from SVD analysis in wild type (**c**) and KO (**d**). **e** Kinetics of SVD1-3 in WT and KO mice (mean value ± SEM; $n = 3$ mice per group). SVD2 shows a small delay in transition through EEs ($^{$$$}p < 0.001$). SVD3 (lysosomal protein degradation) also displays a small right shift ($^{ooo}p < 0.001$). **f, g** Protein uptake length was increased along the proximal convoluted tubule (PCT) in KO mice. Single plane example images (representative of three independent experiments) are depicted 20 min post injection. Scale bars = 20 μm. Histogram depicts the fluorescence signal (mean value ± SD; $n = 3$ mice per group) in ROIs drawn around individual PCT segments, ordered from highest to lowest. Data depicted were from single plane images acquired 20 min post injection of lactoglobulin$^{At565}$. **h, i** Antibody staining for the late endosomal/lysosomal markers Rab7 and Lamp1. Example single plane images are depicted (representative of five independent experiments). Scale bars = 10 μm. **j** Intracellular signal distribution was assessed by the standard deviation (Std. Dev.). In WT mice, signal is condensed in apical vesicles in early PCT segments (high Std. Dev.), and more diffuse in late (low Std. Dev.). This axial pattern was lost in KO mice, resulting in a lack of linear correlation ($R$ value: 0.89 in WT, 0.53 in KO; $p < 0.05$; $n = 5$ mice per group). **k** Summary diagram depicting axial redistribution of protein reabsorption along the PCT in KO mice; increased uptake in later segments can compensate for severe defects in endocytosis.

demonstrates the extraordinarily high degradative capacity of this segment. Finally, fixation methods can have effects on PT cell ultrastructure, and might influence the apparent size of organelles in 3D EM images.

Detailed understanding of how proteins are processed by the kidney in vivo is integral to interpreting the significance of urinary biomarkers. It is widely assumed that the magnitude of tubular proteinuria reflects the severity of protein uptake defects in the PT[7], but we show that compensatory uptake in S2 complicates this relationship. Interestingly, patients with Lowe syndrome display a very high urinary excretion of small peptides (but not free amino acids)[48], which was previously unexplained, but is perhaps consistent with extension of

intact protein uptake into S2, and consequently an effective shortening of the remaining tubular length available for peptide reabsorption. Whether this has consequences for other aspects of downstream S2 function should be investigated in the future. Of note, recent Genome-wide association studies have linked PT endocytotic function to progression of chronic kidney disease[49]. In the meantime, our findings strongly suggest that focusing on intact protein excretion alone provides only a partial assessment of PT function, and that peptides or other markers of protein degradation also need to be considered[50].

## Methods

The research in this study complies with all relevant ethical regulations, and the protocols for animal experiments were approved by The Zurich Cantonal Veterinary Office (ZH194/16). No statistical methods were used to predetermine sample size. The experiments were not randomized, and investigators were not blinded to allocation during experiments and outcome assessment.

### Labeling of proteins and peptides

Bovine $\beta$-lactoglobulin (#L3908 Sigma-Aldrich), recombinant human lysozyme (#L1667 Sigma-Aldrich) and human urine β-2-microglobulin (#126-11 Lee Biosolutions) were labeled with Atto-532 NHS ester (#AD 532-31; Atto-Tec), Atto-565 NHS ester (#AD 565-31; Atto-Tec) or Atto-647N NHS ester (#AD 647N-31; Atto-Tec), according to the manufacturer's protocol. To obtain different labeling densities of $\beta$-lactoglobulin, different dye/protein ratios were used (Supplementary Table 1). Labeled proteins were purified using benchtop PD midiTrap G-25 Sephadex chromatography columns (#28–9180–08; GE Healthcare) and concentrated with 3-kD Amicon Ultra-4 centrifugal filters (#Z740186; Sigma-Aldrich). Concentration was measured by UV/Vis absorption spectroscopy (Libra S70; Biochrom) and labeling density was detected/confirmed with a nano ESI-MS mass spectrometer (Synapt G2 Si, Waters). Labeled proteins were diluted in NaCl 0.9% and 25 ug was injected intravenously in a volume of 120 ul.

To generate peptides, $\beta$-lactoglobulin was trypsin (V5113 Promega) digested in 50 mM ammonium bicarbonate buffer (pH 7.8 37 °C) with a protease: protein ration of 1:100 (w/w). The digest was then analyzed by high pressure liquid chromatography and in-line mass spectrometry (LCMS-2020 Shimadzu, Waters XBridge column C18 3.5 μm 4.6x 250 mm 130 Å, flow 1,0 ml/min, mobile phase A: water 0.1% v/v TFA, B: acetonitrile 0.1% v/v TFA, gradient: 5–10% 5 min, 10–40% 55 min, 40–95% 65 min, 95% 60 min, 95–5% 70 min, 60 °C). The fragments used in the study were identified (mass spectrometry, data analysis by MagTran software, version 1.0), purified and freeze-dried (Speed-Vac SPD120, Thermo Fisher). They were then re-dissolved in the labeling buffer (buffer prepared according the manufacturer's protocol #AD 532-31) and adjusted to pH 8.5 and labeled with Atto-532.

The labeled peptides were again purified by high pressure liquid chromatography and immediately characterized by in-line mass spectrometry. Mobile phase gradients were adapted individually for each peptide (LCMS-2020 Shimadzu, Waters XBridge column C18 3.5 μm 4.6x 250 mm 130 Å, flow 1,0 ml/min, mobile phase A: water 0.1% v/v TFA, B: acetonitrile 0.1% v/v TFA): ALK$^{At532}$−5% 5 min, 5–15% 7 min, 15–45% 55 min, 45–95% 60 min, 95% 68 min, 95–5% 69 min, 95% 70 min; IPAVFK$^{At532}$−5% 5 min, 5–10% 10 min, 10–55% 55 min, 55–95% 60 min, 95% 68 min, 95–5% 69 min, 95% 70 min; GLDIQK$^{At532}$−5–15% 5 min, 15–45% 55 min, 45–95% 60 min, 95% 68 min, 95–5% 69 min, 95% 70 min; VYVEELKPTPEGDLEILLQK$^{At532}$ and VYVEELK$^{At532}$PTPEGDLEILLQK−5% 5 min, 5–30% 5 min, 30–40% 55 min, 40–95% 60 min, 95% 68 min, 95–5% 69 min 95% 70 min.

Exact peptide mass and purity (>95%) were confirmed by LCMS (Waters XBridge column C18 3.5 μm 4.6x 250 mm 130 Å, flow 1.0 ml/min, mobile phase A: water 0.1% v/v TFA, B: acetonitrile 0.1% v/v TFA, gradient: 5% 5 min, 5–95% 20 min, 95% 23 min, 5% 30 min, 60 °C)

(Supplementary Fig. 7). Labeled peptide concentrations were measured using UV/Vis absorption spectroscopy (Libra S70; Biochrom).

The labeling position was confirmed by ESI-MS/MS analysis (Supplementary Fig. 7). Samples were first desalted using C18 Zip Tips (Millipore, USA) and analyzed in MeOH:2-PrOH:0.2% FA (30:20:50). The solution was infused through a fused silica capillary (ID75um) at a flow rate of 1 ul min$^{-1}$ and sprayed through a Pico Tip (ID30um), obtained from New Objective (Woburn, MA). Nano ESI-MS analysis was performed on a Synapt G2_Si mass spectrometer and the data were recorded with the MassLynx 4.2 Software (both Waters, UK). Mass spectra were acquired in the positive-ion mode by scanning the m/z range from 100 to 2000 Th with the scan duration of 1 s and the interscan delay of 0.1 s. The spray voltage was set to 3 kV, the cone voltage to 50 V, and source temperature 80 °C. The species of interest were detected as singly-, doubly-, or triply-charged ions and subjected for structural elucidation by MS/MS.

The peptides were aliquoted and freeze-dried. Labeled peptides were dissolved in NaCl 0.9% and injected intravenously 2.6 nmol in a volume of 75 ul. To assess uptake of the large peptide VYVEELKPTPEGDLEILLQK$^{At532}$ in vitro, PT-derived Opossum Kidney (OK) cells (kind gift from Prof Devuyst, University of Zurich, available from atcc.org: CRL-1840) were used. Cells were incubated for 30 min with the peptide (0.05 μg/μl) and Lysozyme$^{At647N}$ (0.2 μg/μl) at 37 °C. Imaging was performed with a Leica DMI6000B, Model SP8 inverted confocal microscope. The following wavelengths were used: Atto 532 (ex. 532 nm, em. 540–590 nm); Atto 647 N (ex. 647 nm, em 660–710 nm). Image processing was performed using FIJI.

### Characterization of labeled proteins in vitro

All assays were performed on a Synergy 2 plate reader (Biotec) and using 96-well microplates (Greiner Bio-One UV-Star™ 96-Well Microplates). The measurements were performed using an excitation of 540/25 and an 620/40 emission filter. Degradation of labeled proteins ($\beta$-lactoglobulin-Atto 532) in solution was performed by trypsin digestion. First the proteins were diluted from the stock solutions to the concentration 0.2 μg/ul. In each well 190 μl of 50 mM ammonium bicarbonate buffer (pH 7.8) and 10 μl (2.0 μg) of proteins were pipetted. The plates were preincubated at 37 °C, and then the baseline was measured for 30 s. The enzymatic reaction was started by addition of the protease (trypsin 0.4 μg) and then the kinetics were measured for 1 h at 37 °C. A control was measured to account for possible auto-hydrolysis of proteins in the absence of proteases. Protein digestion by cathepsin B (C0150 Sigma-Aldrich) was performed in a buffer consisting of 352 mM Potassium Phosphate, 48 mM Sodium Phosphate, and 4.0 mM Ethylenediaminetetraacetic Acid (pH 6.0) at 40 °C. In each well 180 μl buffer and inhibitor e64 (E3132 Sigma-Aldrich) was added (final concentration 5 μM). Then 5ul cathepsin B (1.0 mg/ml) was pipetted and incubated for 10 min, and 10 μl (2.0 μg) of $\beta$-lactoglobulin was added to each well. To assess protein degradation in cells, OK cells were used, and were dissociated from the culture dish using Trypsin EDTA 0,25% (Gibco™™ Life Technologies, USA). The pellet was washed with PBS and resuspended in buffer (140 mM NaCl, 2.5 mM KCl, 1.8 mM CaCl$_2$, 1 mM MgCl$_2$, 20 mM Hepes). In total, $5 \times 10^6$ cells per well were placed in a 24 well plate (Ibidi 82401), and were incubated with the cathepsin inhibitor e64d (300 μM, Aloxistatin AdipoGen AG), or a broad-spectrum protease inhibitor cocktail (Complete™ ULTRA Tablets, Mini, EDTA-free, EASYpack Protease Inhibitor Cocktail, Roche, Sigma). Lactoglobulin$^{(4xAt532)}$ (1 ug) was added to each well and fluorescence measured with a Synergy 2 plate reader (Biotec) at 37 °C.

### Animals

All experiments were performed on 6–12-week-old male C57Bl/6JRj mice (supplied by Janvier, Le Genest, France), with the exception of experiments using *Clcn5* deficient mice, where females were used

(which have a mosaic expression of the X-linked gene). Mice were housed in the University of Zurich animal facility, at ambient temperature and on a 12-h life cycle, with free access to food (Kliba Nafag formula 3436) and water. They were anaesthetized with 2.5% Isoflurane (Attane, Provet AG, Switzerland) in 100 ml/min Oxygen. The $Ocrl^{y/-}$ mice used in this study were generated previously and represent a humanized model[31]. Knockout of $Ocrl$ alone in mice does not produce a kidney phenotype, due to compensation from INPP5B, a close paralogue. Mice were therefore generated lacking $Ocrl$, but expressing human INPP5B, to provide a humanized background. These mice display a renal tubular phenotype resembling that of humans with Lowe syndrome/Dent 2 disease[31].

### Intravital imaging

The left kidney was externalized for imaging using previously established protocols[20]. The internal jugular vein was cannulated to allow intravenous injections of dyes and reagents. Body temperature was monitored throughout experiments. Imaging was performed on a custom-built multiphoton microscope operating in an inverted mode, using ScanImage software. A broadband tunable laser (InSight DS Dual, Spectraphysics, Santa Clara, USA) was used as excitation source. A × 25 water immersion objective with a numerical aperture of 1.05 and a working distance of 2 mm was used for intravital imaging (Olympus, Tokyo, Japan). The following excitation wavelengths were used: Autofluorescence 850 nm (emission 500–550 nm), Atto-565 850 nm, Atto-532 800 nm and Atto-647N 820 nm. Image processing was performed using FIJI and Imaris version 9.0 (Bitplane AG, Zurich, Switzerland).

### Singular value decomposition analysis

For detailed investigation of intracellular protein processing it was needed to disentangle the spatial and temporal components in the in vivo imaging data. From the EM and fixed tissue immunostaining experiments it was evident that proteins are processed by discreet, clearly localized, but overlapping subcellular structures. This allowed us to treat the time dependent cellular fluorescence intensity distributions as a linear combination of the contributions of the ligands in different cellular compartments. SVD analysis is a model-free approach to decompose such combined signals into the spatial and temporal components of the underlying distinguishable species[23].

In short for a m × n matrix $\mathbf{A}$ the SVD is defined by:

$$\mathbf{A} = \mathbf{USV}^\mathbf{T}$$

$\mathbf{U}$ is an orthonormal m × m matrix; $\mathbf{S}$ is an m × n diagonal matrix and $\mathbf{V}$ is an orthonormal n × n matrix. In the case of this analysis $\mathbf{U}$, also called the left singular vectors of $\mathbf{A}$ contains the spatial distribution of the underlying cellular components. The right singular vector $\mathbf{V}$ contains the kinetic information. The singular values are contained in $\mathbf{S}$ and determine the relative contributions of each pair of columns in $\mathbf{U}$ and $\mathbf{V}$ to $\mathbf{A}$. The singular values are sorted by the magnitude of the contribution to the overall observed signal distribution. For assessing the number of relevant spatial components or base vectors for describing the overall data, we used the amplitudes of the singular values as well as the autocorrelation functions of $\mathbf{U}$ and $\mathbf{V}$.

It is important to note that signals from subcellular components which are spatially separated but share the same or an inverted kinetic behavior can end up in the same base vectors. This means that the number of base vectors or relevant spatial components does not have to be identical to the number of subcellular components. Reconstruction of the overall cellular intensity distribution by setting all singular values but the relevant ones to zero allows to remove non-significant noise contributions and therefore to improve the signals for further interpretation.

To apply this analysis, first the in vivo imaging datasets were rigorously checked for artefacts due to limited imaging stability or due to tissue changes such as collapsing tubules. Next FIJI was used to manually draw 50 line scan ROIs per tubule from the basal to the apical part of PT cells. The temporal intensity profiles were exported for further processing in Wolfram Mathematica. After importing, the x axes of the intensity profiles were rescaled from 0 to 1, followed by mean binning of all profiles of a tubule and per time point. This data was then used for data plotting as well as for the SVD analysis as described above.

### Antibody staining in fixed kidney tissue

Kidney tissue was perfusion fixed via the aorta with 3% paraformaldehyde in a phosphate buffered solution. For line scan experiments and co-staining with antibodies, Atto-565-labeled lysozyme was injected via tail vein injection 1 h prior to fixation, as previously reported[20]. Immune staining was performed on 5 μm thick cryosections incubated overnight with the following antibodies: rabbit anti-Rab11 (D4F5) (1:100, Cell Signaling Technology, #5589), rat anti-Lamp1 (1D4B) (1:100, Abcam, ab25245), mouse anti-Lrp2 (1:200, CD7D5) (Novus Biologicals, NB110-96417), rabbit anti-OAT1 (1:500, Alpha Diagnostic International, OAT11-A), goat anti-Cathepsin L (1:100, R&D Systems, AF1515-SP). The following secondary conjugated antibodies were used (all at 1:500): Dylight488 goat anti-rat (Bethyl Laboratories/ Lubioscience, A110-242D2, a kind gift from Prof Sommer, University of Zurich), Alexa647 donkey anti-rabbit (Jackson ImmunoResearch, 711-606-152), Alexa488 donkey anti-rabbit (Jackson ImmunoResearch, 711-546-152), Alexa647 donkey anti-goat (Jackson ImmunoResearch, 705-606-147), AbberiorStar635-P goat anti-mouse (kind gift from The Center for Microscopy and Image analysis, University of Zurich). Brush-border actin filaments were stained with ActinRed 555 ReadyProbes reagent (Invitrogen, R37112) according to manufacturer's instructions. Images were acquired using a Leica SP8 upright confocal microscope. Line regions of interest were drawn across individual PT segments in cortical labyrinths, and a minimum of 3 mice per marker were analyzed. To assess fluid phase endocytosis, anaesthetized mice were intravenously injected with 3 mg/kg Dextran Alexa Fluor 647. This was followed (after 15 min) by injection of small peptide ALK Atto 532. Twenty minute following the second injection, kidneys were immersion fixed in 3% PFA overnight and processed for paraffin embedding. Paraffin-embedded blocks were sectioned at 5 μm, dewaxed, rehydrated and mounted with Dako mounting medium (Agilent, Santa Clara, California, USA) for imaging.

### Electron microscopy

Mice were perfused (10 ml/min) via the abdominal aorta using a mix of paraformaldehyde (4%) and glutaraldehyde (2.5%) in 0.1 M sodium cacodylate buffer at pH 7.35. After perfusion, the kidney was excised from the mouse and dissected into small (1 mm³) pieces. Samples were rinsed in 0.1 M sodium cacodylate buffer, post-fixed with 1% OsO4 in 0.1 M sodium cacodylate buffer for 1 h at 0 °C, rinsed with $H_2O$ and block-stained with 2% of aqueous uranyl acetate for 1 h at 4 °C, and rinsed with $H_2O$. Samples then were dehydrated in a series of increasing ethanol concentrations in $H_2O$ (70%, 80%, 96%), followed by anhydrous ethanol (100%) and finally propylene oxide, prior to embedding in Epon/Araldite (Sigma-Aldrich, Buchs, Switzerland) and polymerized at 60 °C.

Ultrathin sections (50 nm) were post-stained with Reynolds lead citrate and imaged in a Talos 120 transmission electron microscope (TEM) at 120 kV acceleration voltage equipped with a bottom mounted Ceta camera using MAPS software (Thermo Fisher Scientific, Eindhoven, The Netherlands). Tile sets were used to identify and select the region of interest on the resin block for subsequent FIB-SEM.

### Focused ion beam scanning electron tomography

The selected Epon/Araldite block was mounted on a regular SEM stub using conductive carbon and coated with 10 nm of carbon by electron beam evaporation to render the sample conductive. Ion milling and image acquisition was performed simultaneously in an Auriga 40 Crossbeam system (Zeiss, Oberkochen, Germany) using the FIBICS Nanopatterning engine (Fibics Inc., Ottawa, Canada). A large trench was milled at a current of 16 nA and 30 kV, followed by fine milling at 240 pA and 30 kV during image acquisition with an advance of 5 nm per image. SEM images were acquired at 1.9 kV (30 μm aperture) using an in-lens energy selective backscattered electron detector with a grid voltage of 550 V, a dwell time of 1 μs, and a line averaging of 40 lines. The pixel size was set to 5 nm and tilt-corrected to obtain isotropic voxels. The final image stack was registered and cropped to the area of interest using the Fiji image-processing package[51]. Segmentation of ELS structures was performed using Ilastik 1.0[52].

### Statistical analyses

For comparison of uptake capacity in animal models of disease (*Ocrl* and *Clcn5*), two-way ANOVA repeated measures was applied on the first 10 min from injection, and main effect of genotype was evaluated. All kinetic datasets were analyzed using two-way ANOVA repeated measures and the treatment (or genotype) × time interaction significant datasets are indicated in the manuscript and *p* value is reported.

### Reporting summary

Further information on research design is available in the Nature Research Reporting Summary linked to this article.

## Data availability

Source data are provided with this paper.

## Code availability

The code used in this study is available at a public repository (https://osf.io/qfzdt/?view_only=4a5d0b4d6bd847278a3d4cca95b1f00b).

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

## Acknowledgements

The authors acknowledge support from The Center for Microscopy and Image Analysis, University of Zurich, and The Functional Genomics Center Zurich. The authors are grateful to Prof Olivier Devuyst, University of Zurich, for providing *Ocrl* and *Clcn5* deficient mice. Funding was from The Swiss National Centre for Competence in Research (NCCR) Kidney Control of Homeostasis and The Swiss National Science Foundation (310030_184688) to A.M.H.

## Author contributions

M.P., M.B. (Bugarski), C.S. and N.J. performed the in vivo experiments and data analysis. M.K. was responsible for the generation and characterization of labeled proteins/peptides, with assistance from D.H. D.H. designed and implemented the computational analysis approach. M.P., A.K. and J.M.M. performed the FIB-SEM experiments and 3D reconstruction. M.B. (Berquez) generated and provided the knockout mice. A.M.H. supervised the project and wrote the manuscript with input from all authors.

## Competing interests

The authors declare no competing interests.
