## [Peer Review File · Nature Communications]

REVIEWER COMMENTS

Reviewer #1 (Remarks to the Author):

In this manuscript, Polesel et al. describe a series of studies that combine intravital microscopy, electron microscopy and a sophisticated mathematical analysis to characterize processing of luminal proteins by the proximal tubule of mice.

The investigators have made clever use of fluorogenic reporters in time-series intravital microscopy studies of the kidneys of mice, allowing them to distinguish uptake and degradation of endocytic substrates by the proximal tubule in a way that provides compelling evidence of sequential protein processing by S1 and S2 segments. By labeling different endocytic ligands with fluorophores at stoichiometries where self-quenching occurs, their degradation following internalization can be detected by an increase in fluorescence. While, in principle, it can be difficult to distinguish increases in fluorescence arising from unquenching from increases arising from internalization and fusion of endocytic vesicles, the authors present the results of a very rigorous set of studies that convincingly demonstrate their ability to distinguish the processes of uptake and unquenching. These studies include exhaustive characterization of the probes, results of the effects of competitive inhibitors of uptake and lysosomal hydrolysis and impressive studies of endocytic processing of probe degradation products. The result of these studies is an exciting, novel and compelling model of sequential endocytic processing of filtered material by different segments of the proximal tubule.

However, the second part of the manuscript describes the use of a “singular value decomposition” (SVD) analysis of axial line scans of proximal tubule fluorescence that is interpreted with respect to endocytic compartments as identified by electron microscopy. This analysis is unconvincing and is used to generate a model of transport that is both ornate and speculative. SVD is a relatively arcane mathematical procedure that may or may not be appropriate to the studies shown here. While the authors describe the SVD analysis, they do not describe it well enough to help a reader understand it from first principles, nor do they give any examples where SVD has been applied to analyze systems like those analyzed here. The authors reference a paper published in 1972, that describes SVD as a mathematical construct, with a few examples of its application to spectroscopy and reaction kinetics. While intriguing, the authors provide no evidence of the validity of the results (e.g., analyses of data collected at different time points, analysis of an established model system, demonstration that three components are necessary and sufficient).

The three SVD “base vectors” derived from this analysis are interpreted as functionally distinct compartments that are mapped to endocytic compartments (early endosomes, large apical vacuoles and lysosomes) as identified by electron microscopy. The problem is that while specific functions are assigned to these different compartments, they are identified only by morphology. In the absence of any

functional information about these compartments (for example the kinetics of endocytic access, cargo contents or association with diagnostic compartmental markers such as particular Rab proteins), the mapping of the SVD base vectors to specific endocytic compartments seems completely speculative. Consequently, it is difficult to take the complex models of trafficking such as those shown in Figure 5 and Supplementary Figure 6 seriously. Among the essentially unsupported conclusions that the authors derive from this analysis are that the studies demonstrate that material moves from lysosomes back to LAVs, that degradation occurs only in lysosomes and that LAVs play a role in sorting.

Insofar as the electron microscopy is inadequate to provide a map of the functionally distinct compartments corresponding to the SVD base vectors, it is not clear that it adds value to the manuscript. In addition to providing questionable compartmental context for the SVD analysis, the authors frequently over-reach in their interpretation of the images. One image shows tubular structures extending from a compartment that are identified as “budding of a recycling tubule”. Another image shows a set of tubules that are identified as a “network of recycling tubules” A third shows one ill-defined structure that is identified as an “LAV wrapping around Lys”. At other points, the authors claim that their data demonstrate connections between LAVs and the TGN (apparently identified by morphology alone). The authors should also explain the basis for the colorization of the electron micrographs shown in Figure 4.

The same approaches are then extended to studies of two models of disease, OCRL and Clcn5 knockout mice. While the studies of the OCRL mouse nicely demonstrate how a defect in S1 endocytosis results in enhanced uptake by S2, I am not convinced that this effect should be interpreted as “compensatory remodeling”, as is frequently mentioned. All of the same caveats discussed above apply to the SVD analyses of these studies, which appear to provide no new insights.

Finally, given the fundamental importance of distinguishing S1 from S2 segments in these studies the authors should consider a more complete explanation of how they distinguish the two. The image shown in Figure 1A does not inspire confidence and raises the question of how green probe fluorescence is distinguished from autofluorescence. While the immunofluorescence shown in Figure 2 and Supplementary Figure 4 are the most compelling, an association of distinctive patterns of autofluorescence with a time series of a fluorescent probe transiting the tubule would suffice.

Reviewer #2 (Remarks to the Author):

This ambitious manuscript is broadly focused on the uptake and processing of filtered proteins by the kidney proximal tubule. The authors use state-of-the-art imaging and computational approaches to

follow the internalization and degradation of fluorescently-labeled lactoglobulin, other proteins, and peptides of varying sizes along the tubule axis. They posit three major conclusions- that “complete degradation of proteins requires sequential coordinated activity of distinct PT sub-segments, each displaying genetic and structural adaptations to specific tasks, implying that protein metabolism shapes the axial topography of the PT”; that “the kinetics of renal protein processing are remarkably consistent between animals, suggesting a highly conserved mechanism”; and that “compensatory remodelling of the PT can limit urinary protein loss” in disease models.

The experiments are elegantly conceived, and many of the images are spectacular. The authors attempt to present an enormous body of work within the tight constraints of the journal. I am very appreciative of the effort and expertise required to perform this work, and excited about the potential significance of these studies to our understanding of tubular function. I do have significant concerns about the interpretation of some of the data, particularly the data in Figs 1-3. The authors also make numerous unsubstantiated sweeping conclusions that are not supported by their data and in some instances appear inconsistent with their previously published studies. The work would have more significant and long-lasting impact if there were less of an emphasis on perceived novelty in favor of a more balanced treatment of the limitations of these studies and a broader discussion of the implications for normal and disease physiology.

A major issue throughout is the lack of experimental detail about how the experiments were performed and the imaging conducted. The figure legends should clearly state n values, time courses for imaging, and what is shown- are these single planes or image stacks (and if so, are they max or sum projections)? What objectives were used and were images processed identically? What wavelength is used to visualize autofluorescence? Omission of these variables (and particularly time and imaging conditions) makes it difficult to evaluate the authors’ conclusions.

The first few figures show an increase in fluorescence in S1 and appearance in S2 segments in mice injected with fluorescent lactoglobulin or b2 microglobulin and imaged after 10 vs 100-120 min. Inclusion of a cathepsin inhibitor reduces the fluorescence, confirming that the increase is due to dequenching of degraded ligand. Subsequently, the authors inject fluorescent trypsinized fragments into mice and find that a longer (~20 amino acid) fragment appears preferentially in S1, together with lysozyme, whereas a tripeptide is taken up primarily in S2 segments. They make the curious statement that this implies the “existence of a size threshold, above which megalin mediated endocytosis occurs”.

The authors then graph the distribution of selected receptors and peptidase transcripts in each subsegment taken from Knepper’s microdissected mouse PT dataset. Because a subset of peptidases and the peptide transporter PEPT2 are preferentially enriched in S2 and S3 segments, they conclude that the delayed appearance of fluorescence in S2 segments represents peptide transporter-mediated uptake of degraded peptide fragments released apically from S1 cells.

I believe there are more parsimonious explanations for this collection of results that are still of significant interest. The interpretation that the appearance of fluorophore in S2 is due to uptake of released degraded fluorescent peptides released from S1 cells that have taken up and processed intact ligand is predicated on the authors’ assertion that filtration of the injected ligand is fully complete within 10 min of injection, a conclusion that is never verified and which also seems inconsistent with the kinetics of distal tubule levels of free dye shown in Fig. S1. The authors could potentially support their claim by measuring the reduction in plasma fluorescence or by showing that there is no change in S1

fluorescence intensity over long periods when the nonquenched sparsely conjugated fluorophore is injected in the presence of cathepsin inhibitors. However, it seems to me far more likely that the appearance of tubular fluorescence in S2 represents the gradual uptake and processing of intact protein over the 2 h time course of imaging. S2 segments express high levels of megalin and cubilin receptors, and all modeling studies describing the uptake of albumin and other ligands conclude that residual uptake does occur beyond the S1 segment. Indeed, the authors confirm that this is the case in Figure S3 by showing that injection of lysine results in a shift in axial reabsorption of lactoglobulin to the S2 segment.

With respect to the conclusion that the uptake of small fragments into S2 cells occurs via peptide transporters, the authors themselves have shown in previous elegant work that uptake of the fluid phase marker dextran is robust in the S2 segment relative to S1 (Schuh et al), a finding that was recently confirmed (Edwards et al, PMID 35178707). What evidence is there that uptake of their tripeptide does not occur by fluid phase uptake? Use of a transporter knockout model could address this issue. The preferential uptake of their larger peptide in S1 might indicate some affinity of this fragment for megalin or cubilin, a finding that could be readily tested in cell culture or by SPR.

The RNASeq data in Figure 3 are chosen to support the authors' preferred conclusions, but are not entirely congruent with quantitative proteomic data from the same source. Based on the proteomic dataset (esbl.nhlbi.nih.gov/KTEA/) megalin levels are doubled in S2 vs S1 and PEPT2 levels are ~5x higher in S3 vs S2. Acknowledgment of the proteomic data and/or the limitations of transcriptomic data in making conclusions about protein expression levels should be noted.

The authors suggest with no evidence in their schematic in Fig 3C that fragment uptake occurs gradually in S2 and is maximal in S3. They cannot image this segment intravitaly, and it is impossible to ascertain from the data in Fig 2I whether there is any uptake in S3 segments. Because S3 segments have high peptide transporter levels but very low expression of Dab2, staining fixed kidneys similar to those in Fig 2 with markers to directly compare tripeptide vs dextran uptake in S2 vs S3 segments would enable the authors to distinguish between fluid phase vs transporter-mediated uptake.

The authors suggest that lysine competes for megalin binding, but the role of effect of lysine on protein reabsorption in mice is irreversible and likely complex (see for example Sumpio and Maack, *AJP-Renal* (1982) 243:F379) Additionally, the time course for imaging in Fig S3b is not noted. If lactoglobulin is fully filtered within 10 min then surely all the lysine must be gone as well? Conversely, if lysine were in fact competing directly for uptake, then the differential conjugation of accessible lysine residues in their lactoglobulin probes may be problematic. The SVD1 kinetics showing differential uptake of low and highly conjugated lactoglobulin in Fig 5i and j would appear to bear this out.

The authors note that a fraction of degraded peptides processed by the PT is known to be released lumenally, however, a considerable fraction is also released basolaterally. One would predict that this fluorescence would be somewhat concentrated and visible within the interstitium, but this does not seem to be the case.

The FIB-SEM images are spectacular and clearly required an enormous effort. A surprising feature is the large apical invaginations, which look very different from those observed in conventional EM images of the proximal tubule. Please provide the fixation and perfusion rate conditions for how the tissue was

prepared and comment on whether this might represent an artifact of the tissue preparation or whether this feature is obscured in conventional EM.

How the FIB-SEM data are correlated to the SVD shapes and what these shapes represent in subsequent figures is uninterpretable as described. What is the x axis in Fig 3f and g? Why is low-labeled lactoglobulin visible immediately upon injection whereas high-labeled lactoglobulin is visible only after reaching lysosomes? The characterization of these probes in supplemental data shows that the high-labeled version has comparable or higher fluorescence intensity at physiologic pH per molecule.

The grandiose conclusion in the abstract that “the kinetics of renal protein processing are remarkably consistent between animals, suggesting a highly conserved mechanism” means only that three (male) mice showed similar results. It seems premature to state based on this that ELS dynamics in vivo “are far less heterogeneous than in non-specialised cell culture models”.

The Lowe Syndrome mouse model is described as *Ocrly*^{-/-} but based on the reference provided and the lack of a renal phenotype in *Ocr1* knockouts, this is likely the humanized mouse model that overexpresses human INPP5B in place of the endogenous mouse isoform. The model should be mentioned in the methods sections and potential limitations of this model should be noted.

The concept that there is axial “remodeling” in *Ocr1* knockout mice implies a functional change in segment identity or in segment length, a finding that is unsupported by the data presented. Rather, their data suggest that a reduction in megalin expression in these mice causes an increase in the axial length across which protein uptake occurs. In fact, this prediction has previously been suggested in a published model that is not curiously not referenced in the study (Gliozzi et al. (2020) JASN 31:67).

Do the authors have data from autofluorescence measurements and morphometry to suggest any change in cortical segment length? The study referenced above quantified shorter S1 segments in zebrafish *Ocr1* knockouts. Is the “remodeling” that the authors propose here congruent with this finding?

Data from the *Clc5* heterozygote knockout mice are challenging to interpret. The “x-” ROIs marked in Fig 6B appear to be in a region where the tubule is curving away from the plane shown. More details about what was done might clarify this point.

Minor issues:

The title and abstract do not support the conclusions

The authors suggest in the introduction that ligand binding to megalin and cubilin triggers endocytosis, whereas, the general consensus is that trafficking of these receptors is constitutive. If the authors have evidence that this is the case, it should be cited.

There is a disconnect between the references cited in the text and the list at the end.

S2 segments in Fig 1c,e,h,and j should be indicated. Tubules in Fig 1k appear significantly dilated compared with the same section in 1j. Is this evidence of renal distress at this late time point?

The beginning of the results section is not specified.

There are occasional typos in the figures

There are some clunky labels- eg, “surface reached by the primary urine” in Fig 4b.

What do the boxes in Fig 4a represent? They appear to be largely luminal.

The suggestion that there are some “potential limitations” to the approaches seems understated at best.

A significant obstacle in this field is the challenge of obtaining quantitative in vivo data, and this team has enviable expertise in this area that enables them to address questions that can be studied by a handful of groups worldwide at best. Although science is ultimately self-correcting, these investigators have a particular obligation to objectively evaluate the strengths and limitations of their studies, specifically because they are so difficult to perform, replicate, and extend. A more balanced treatment of the data presented here, including a tighter focus on the FIB-SEM and subsequent figures, would considerably enhance the value of this work.

Reviewer #3 (Remarks to the Author):

This manuscript from Polesel et al describes the labelling of proteins and peptides with fluorescent dyes in order to image their progression through the proximal tubules of living mice. The authors use Singular Value Decomposition (SVD) analysis to deconvolute spatially and temporally composite signals and identify a sequential activity of distinct proximal tubule sub-segments for protein degradation.

The results are interesting; however, a large amount of data is presented, and the explanation of the data presented in the figures in the main text is very brief. Clarity and readability of the data would be improved in a longer form article, i.e. a research article rather than a communication. The discussion is in general well-reasoned, however there are concerns to be addressed.

- The statement in the abstract (also in the main text): “We also find that the kinetics of renal protein processing are remarkably consistent between animals, suggesting a highly conserved mechanism” should be reworded as it can be interpreted that different species of animals have been investigated, which is not the case.
- The authors refer to “sophisticated computational analysis” in the abstract and elsewhere, however more specific reference to SVD analysis is preferable.

Regarding the generation and characterisation of labelled proteins and peptides:

- The number of investigated peptides (purified and labelled) does not appear to be stated in the text or supplementary information. The methods state that peptides were individually purified and characterised: “Exact peptide mass and purity (>95%) were confirmed” and “The labelling position was confirmed by ESI-MS/MS analysis.” A summary of these data and the associated spectra and chromatograms should be provided in the supplementary information.

- The methods also state: “Mobile phase gradients were adapted individually for each peptide.” However, this information is not given. Reproducibility would be improved were this to be included in a summary table of identified peptides.

REVIEWER COMMENTS

We thank the reviewers for their helpful comments on our manuscript. A number of the issues raised below from all 3 reviewers essentially relate to the brevity and style of the original article. By way of explanation, this manuscript was originally submitted to another member of the Nature journal family that has very tight space limits, which restricted our ability to explain the work in more detail. It was subsequently redirected to Nature Communications, where the larger space allowance has enabled us to substantially increase the amount of data, information, explanation and discussion provided in the revised version, including greater acknowledgement of the limitations of some of the approaches. We hope this will satisfy the concerns of the reviewers. We apologize that this information was not clear enough in the original version and thank the reviewers for their understanding.

Reviewer #1 (Remarks to the Author):

In this manuscript, Polesel et al. describe a series of studies that combine intravital microscopy, electron microscopy and a sophisticated mathematical analysis to characterize processing of luminal proteins by the proximal tubule of mice.

The investigators have made clever use of fluorogenic reporters in time-series intravital microscopy studies of the kidneys of mice, allowing them to distinguish uptake and degradation of endocytic substrates by the proximal tubule in a way that provides compelling evidence of sequential protein processing by S1 and S2 segments. By labeling different endocytic ligands with fluorophores at stoichiometries where self-quenching occurs, their degradation following internalization can be detected by an increase in fluorescence. While, in principle, it can be difficult to distinguish increases in fluorescence arising from unquenching from increases arising from internalization and fusion of endocytic vesicles, the authors present the results of a very rigorous set of studies that convincingly demonstrate their ability to distinguish the processes of uptake and unquenching. These studies include exhaustive characterization of the probes, results of the effects of competitive inhibitors of uptake and lysosomal hydrolysis and impressive studies of endocytic processing of probe degradation products. The result of these studies is an exciting, novel and compelling model of sequential endocytic processing of filtered material by different segments of the proximal tubule.

We thank the reviewer for their positive comments, which are much appreciated.

However, the second part of the manuscript describes the use of a “singular value decomposition” (SVD) analysis of axial line scans of proximal tubule fluorescence that is interpreted with respect to endocytic compartments as identified by electron microscopy. This analysis is unconvincing and is used to generate a model of transport that is both ornate and speculative. SVD is a relatively arcane mathematical procedure that may or may not be appropriate to the studies shown here. While the authors describe the SVD analysis, they do not describe it well enough to help a reader understand it from first principles, nor do they give any examples where SVD has been applied to analyze systems like those analyzed here. The authors reference a paper published in 1972, that describes SVD as a mathematical construct, with a few examples of its application to spectroscopy and reaction kinetics. While intriguing, the authors provide no evidence of the validity of the results (e.g.,

analyses of data collected at different time points, analysis of an established model system, demonstration that three components are necessary and sufficient).

SVD analysis is a well-established mathematical approach that allows disentangling of overlapping signals in an unbiased and model-free manner. It has been used widely with various different imaging modalities, including fluorescence microscopy (PMID: 10769049, PMID: 28103588), CT (PMID: 29762256), MRI (PMID: 25291148), PET (PMID: 27567671), EM (PMID: 33032160) and photoacoustic imaging (PMID: 28101402). We apologize that more recent references were missing from the original manuscript (due to space constraints) – these have now been added. We have also added additional information concerning how the analysis was performed in this case. Again, we are sorry this was not clearer before. We are not aware of any studies using SVD analysis to investigate the endo-lysosomal system (ELS), but our specific reasons for doing so are explained below. We agree that application to an established model system would be ideal, but we are unaware of any model that recreates the structure/function of the PT ELS *in vivo*. We have also provided a clearer justification in the revised manuscript as to why 3 components are sufficient to almost completely recreate the raw data.

The three SVD “base vectors” derived from this analysis are interpreted as functionally distinct compartments that are mapped to endocytic compartments (early endosomes, large apical vacuoles and lysosomes) as identified by electron microscopy. The problem is that while specific functions are assigned to these different compartments, they are identified only by morphology. In the absence of any functional information about these compartments (for example the kinetics of endocytic access, cargo contents or association with diagnostic compartmental markers such as particular Rab proteins), the mapping of the SVD base vectors to specific endocytic compartments seems completely speculative. Consequently, it is difficult to take the complex models of trafficking such as those shown in Figure 5 and Supplementary Figure 6 seriously. Among the essentially unsupported conclusions that the authors derive from this analysis are that the studies demonstrate that material moves from lysosomes back to LAVs, that degradation occurs only in lysosomes and that LAVs play a role in sorting.

Investigating ELS function in the proximal tubule (PT) represents a substantial technical challenge, due to: (1) the lack of representative *in vitro* models; (2) the paucity of established markers to label specific components, (3) the close proximity and spatial overlap of individual structures. These long-standing problems provided the motivation for us to take the approach described in the paper as a first attempt to move the field forward, and we reasoned that SVD analysis might provide a suitable method to circumvent some of these hurdles (i.e. disentangling overlapping processes, no pre-requisite for an existing model). Our analysis, which was highly reproducible, strongly suggested the existence of 3 major processes, which together almost completely recreate the dynamics of protein handling in the ELS *in vivo*.

The spatial profiles of these 3 base vectors were highly suggestive that they each have a structural basis. However, the diagnostic compartmental markers mentioned by the reviewer are unfortunately not currently available *in vivo* (indeed, had we such information it is questionable that SVD analysis would be necessary).

We therefore used characteristic structural features – derived from 3D EM images and antibody staining – to assign probable identities to the SVD base vectors. These were then validated by interventions (including cathepsin inhibition and usage of quenched/non-quenched proteins) that showed expected

responses according to our model. For example, inhibition of lysosomal cathepsins abolished SVD vector 3, implying that it reflects protein catabolism. Moreover, we were able to demonstrate that SVD vector 1 has two distinct time components, the second of which is also dependent on protein degradation. To further strengthen these conclusions, we have performed new antibody staining for lysosomal cathepsin L in fixed kidney tissue (Fig 5k), to better demonstrate that fluorescence unquenching of highly labeled proteins in live imaging studies occurs in small vesicular structures resembling the structure and location of lysosomes.

However, we acknowledge that the model may be oversimplified, and we cannot exclude the possibility that it may be missing some nuance (e.g. sub-classes of different organelles), especially as further specific interventions are very difficult to perform in vivo. We have therefore clearly acknowledged the limitations in the revised manuscript. Moreover, we have reworded the results section to state that these are probable identities, which might be better conceptualized as spatially localized processes, rather than defined structures.

Despite the limitations of the approach, overall, we believe it fits well with knowledge of the architecture of the PT ELS (PMID: 27813828). Moreover, our kinetic measurements are in line with older autoradiographic studies, where the ELS components could be precisely identified (PMID: 186659; PMID: 8476017). However, such studies could only be performed using single EM images from fixed tissue acquired from different animals; the major advantage of our approach is that such kinetic measurements can now be made in real time in the same living mice (e.g. to identify abnormalities in disease models).

In summary, we understand the concerns of the reviewer, and have stated the limitations of the analysis more clearly in the revised manuscript, but remain of the opinion that it provides a useful working model (e.g. to characterize defects in KO mice), which can be revised accordingly in the future as technology develops.

Insofar as the electron microscopy is inadequate to provide a map of the functionally distinct compartments corresponding to the SVD base vectors, it is not clear that it adds value to the manuscript. In addition to providing questionable compartmental context for the SVD analysis, the authors frequently over-reach in their interpretation of the images. One image shows tubular structures extending from a compartment that are identified as “budding of a recycling tubule”. Another image shows a set of tubules that are identified as a “network of recycling tubules”. A third shows one ill-defined structure that is identified as an “LAV wrapping around Lys”. At other points, the authors claim that their data demonstrate connections between LAVs and the TGN (apparently identified by morphology alone). The authors should also explain the basis for the colorization of the electron micrographs shown in Figure 4.

Our motivation for performing FIB-SEM was to evaluate the 3-D topography of the PT ELS. This allowed segmentation of different structures, denoted by colors (this information has been added to the legend). 3-D analysis yielded several useful insights for interpreting the live imaging data. For example, it revealed that LAVs are considerably larger and more irregular than adjacent lysosomes, which was not so obvious from 2D images alone. Moreover, along with antibody staining it reinforced the existence of overlapping functional zones in the PT ELS. Nevertheless, we understand the point that the reviewer is making, and have shortened and reworded this section, with a focus purely on concepts that are central

to the main topics of the paper. Given the apparent return of protein fragments from the LAVs/lysosomes to the tubular lumen in S1, we thought the existence of numerous recycling tubules was worth highlighting (also demonstrated in previous studies, which were referenced in the manuscript).

The same approaches are then extended to studies of two models of disease, OCRL and Clcn5 knockout mice. While the studies of the OCRL mouse nicely demonstrate how a defect in S1 endocytosis results in enhanced uptake by S2, I am not convinced that this effect should be interpreted as “compensatory remodeling”, as is frequently mentioned. All of the same caveats discussed above apply to the SVD analyses of these studies, which appear to provide no new insights.

We appreciate the point that the reviewer is making regarding terminology, and have replaced the term “compensatory remodeling” with “compensatory uptake”. We respectfully disagree that the analysis did not provide new insights. For example, we were able to show evidence that the primary cellular defect in OCRL KO mice is in endocytosis, rather than downstream trafficking or lysosomal degradation. To the best of our knowledge, this was not at all clear from previous *in vivo* or *in vitro* studies. Moreover, we provide evidence of clear divergence in ELS phenotype *in vivo* between OCRL and CLC5 KO mice. Again, we are not aware that this has been previously reported, but it helps to explain known differences in magnitude of proteinuria.

Finally, given the fundamental importance of distinguishing S1 from S2 segments in these studies the authors should consider a more complete explanation of how they distinguish the two. The image shown in Figure 1A does not inspire confidence and raises the question of how green probe fluorescence is distinguished from autofluorescence. While the immunofluorescence shown in Figure 2 and Supplementary Figure 4 are the most compelling, an association of distinctive patterns of autofluorescence with a time series of a fluorescent probe transiting the tubule would suffice.

We have previously described differences in S1/S2 autofluorescence (AF) signals in detail (e.g. see PMID: 30132348, PMID: 30301861). This approach has also been used to identify PT segments by other groups (e.g. see PMID: 21784899). These AF signals are strongest in the green range. For experiments with lactoglobulin labeled with Atto-565, we were able to co-image the red 565 signal with green AF (please see example in SF1). For experiments with lactoglobulin labeled with Atto-532 (which is nearer the green range), tubules were also first identified with AF, but the laser power was then lowered to remove the background signal before protein injection. Thus, it is not possible to co-image autofluorescence and protein signals. However, we have provided an example in SF1 with a sum image of the background AF signal (i.e. the first few frames pre-injection), and the subsequent appearance of Atto532 signals in identified segments with time.

Reviewer #2 (Remarks to the Author):

This ambitious manuscript is broadly focused on the uptake and processing of filtered proteins by the kidney proximal tubule. The authors use state-of-the-art imaging and computational approaches to follow the internalization and degradation of fluorescently-labeled lactoglobulin, other proteins, and peptides of varying sizes along the tubule axis. They posit three major conclusions- that “complete degradation of proteins requires sequential coordinated activity of distinct PT sub-segments, each

displaying genetic and structural adaptations to specific tasks, implying that protein metabolism shapes the axial topography of the PT"; that "the kinetics of renal protein processing are remarkably consistent between animals, suggesting a highly conserved mechanism"; and that "compensatory remodelling of the PT can limit urinary protein loss" in disease models.

The experiments are elegantly conceived, and many of the images are spectacular. The authors attempt to present an enormous body of work within the tight constraints of the journal. I am very appreciative of the effort and expertise required to perform this work, and excited about the potential significance of these studies to our understanding of tubular function. I do have significant concerns about the interpretation of some of the data, particularly the data in Figs 1-3. The authors also make numerous unsubstantiated sweeping conclusions that are not supported by their data and in some instances appear inconsistent with their previously published studies. The work would have more significant and long-lasting impact if there were less of an emphasis on perceived novelty in favor of a more balanced treatment of the limitations of these studies and a broader discussion of the implications for normal and disease physiology.

We thank the reviewer for their positive comments, which are much appreciated. Regarding the style of the original submission, please see our general comments above. We have now utilized the extra space allowance of the current journal to provide more explanation and discussion in the revised version. We apologize that this was missing originally and thank the reviewer for their understanding.

A major issue throughout is the lack of experimental detail about how the experiments were performed and the imaging conducted. The figure legends should clearly state n values, time courses for imaging, and what is shown- are these single planes or image stacks (and if so, are they max or sum projections)? What objectives were used and were images processed identically? What wavelength is used to visualize autofluorescence? Omission of these variables (and particularly time and imaging conditions) makes it difficult to evaluate the authors' conclusions.

We apologize that this information was not clear and have updated the figure legends accordingly. Single image planes are shown throughout, with the exception (where denoted) of the usage of sum images. The autofluorescence signals were excited at 850nm. A $\times 25$ water immersion objective with a numerical aperture of 1.05 and a working distance of 2 mm was used for intravital imaging (Olympus, Tokyo, Japan) – this information has been added to the methods.

The first few figures show an increase in fluorescence in S1 and appearance in S2 segments in mice injected with fluorescent lactoglobulin or b2 microglobulin and imaged after 10 vs 100-120 min. Inclusion of a cathepsin inhibitor reduces the fluorescence, confirming that the increase is due to dequenching of degraded ligand. Subsequently, the authors inject fluorescent trypsinized fragments into mice and find that a longer (~20 amino acid) fragment appears preferentially in S1, together with lysozyme, whereas a tripeptide is taken up primarily in S2 segments. They make the curious statement that this implies the "existence of a size threshold, above which megalin mediated endocytosis occurs". The authors then graph the distribution of selected receptors and peptidase transcripts in each subsegment taken from Knepper's microdissected mouse PT dataset. Because a subset of peptidases and the peptide transporter PEPT2 are preferentially enriched in S2 and S3 segments, they conclude that

the delayed appearance of fluorescence in S2 segments represents peptide transporter-mediated uptake of degraded peptide fragments released apically from S1 cells.

We observed that filtered large (20aa) peptides were taken up in S1, whereas smaller (3 and 6aa) fragments passed straight through to S2. These results suggest that at a certain size, peptides can trigger uptake in S1, hence the statement "...existence of a size threshold, above which megalin mediated endocytosis occurs". However, since we have not investigated the nature of this threshold in detail, we have removed this sentence.

I believe there are more parsimonious explanations for this collection of results that are still of significant interest. The interpretation that the appearance of fluorophore in S2 is due to uptake of released degraded fluorescent peptides released from S1 cells that have taken up and processed intact ligand is predicated on the authors' assertion that filtration of the injected ligand is fully complete within 10 min of injection, a conclusion that is never verified and which also seems inconsistent with the kinetics of distal tubule levels of free dye shown in Fig. S1. The authors could potentially support their claim by measuring the reduction in plasma fluorescence or by showing that there is no change in S1 fluorescence intensity over long periods when the nonquenched sparsely conjugated fluorophore is injected in the presence of cathepsin inhibitors. However, it seems to me far more likely that the appearance of tubular fluorescence in S2 represents the gradual uptake and processing of intact protein over the 2 h time course of imaging. S2 segments express high levels of megalin and cubilin receptors, and all modeling studies describing the uptake of albumin and other ligands conclude that residual uptake does occur beyond the S1 segment. Indeed, the authors confirm that this is the case in Figure S3 by showing that injection of lysine results in a shift in axial reabsorption of lactoglobulin to the S2 segment.

We understand the point that the reviewer is making, and agree that S2 cells do have capacity for protein uptake (as demonstrated by lysine and OCRL KO experiments). However, under control conditions, we do not observe lactoglobulin reaching S2 cells, due to the very high uptake capacity of S1, which we have demonstrated previously far exceeds the normal filtered load (PMID: 30301861). In the kinetic experiments in F1g it can be observed that fluorescence signal only becomes visible in S2 segments after a very long delay (>20 minutes), and after the onset of protein degradation in S1. In contrast, we have seen previously that direct uptake of substrates in S2 (e.g. filtered dextrans) is clearly visible within 1 minute of intravenous injection. Moreover, if there was delayed filtration of small amounts of intact proteins, it is not clear to us how this would bypass the high uptake capacity of S1. We have previously performed experiments with sequential injections of proteins, and have found no evidence that S1 uptake becomes saturated. As suggested by the reviewer, we have now included data on the vascular fluorescence signal (SFig 1), where it can be appreciated that filtration is almost complete within 10 minutes.

In summary, for the reasons given we think it unlikely that uptake in S2 simply reflects delayed filtration, but to demonstrate our openness to alternative explanations we have mentioned this possibility in the revised discussion, for the benefit of readers. Of note, in the original version of the manuscript we referenced older studies describing that albumin fragments can be detected in urine, consistent with release from the PT post lysosomal degradation. During the course of preparing this response, we also became aware of another nice study using mass spectrometry imaging in mouse kidney sections

reporting the existence of albumin fragments in tubular lumens, which were predicted to arise from cathepsin activity (PMID: 29779708). This study has now been referenced.

With respect to the conclusion that the uptake of small fragments into S2 cells occurs via peptide transporters, the authors themselves have shown in previous elegant work that uptake of the fluid phase marker dextran is robust in the S2 segment relative to S1 (Schuh et al), a finding that was recently confirmed (Edwards et al, PMID 35178707). What evidence is there that uptake of their tripeptide does not occur by fluid phase uptake? Use of a transporter knockout model could address this issue. The preferential uptake of their larger peptide in S1 might indicate some affinity of this fragment for megalin or cubilin, a finding that could be readily tested in cell culture or by SPR.

We agree with the reviewer that macromolecules with lower affinity for S1 uptake than intact proteins (e.g. dextrans) can reach S2 and be reabsorbed by non-specific fluid phase endocytosis. However, in our experience substantial fluid phase endocytosis also occurs in S1. In contrast, we observed almost no S1 uptake of small peptide fragments, which was for us a highly intriguing finding. Moreover, we also observed very high uptake of fragments in S3, where would not expect much fluid phase endocytosis to occur (please see below). To demonstrate this point, we have performed new experiments in mice injected with dextran and peptide (SFig 4), which display very different uptake patterns along the PT. Moreover, as suggested, we have also included new data on large peptide uptake in PT-derived (OK) cells (SFig 4), where co-localization with intact protein (lysozyme) in endo-lysosomal vesicles can be appreciated.

The RNASeq data in Figure 3 are chosen to support the authors' preferred conclusions, but are not entirely congruent with quantitative proteomic data from the same source. Based on the proteomic dataset (esbl.nhlbi.nih.gov/KTEA/) megalin levels are doubled in S2 vs S1 and PEPT2 levels are ~5x higher in S3 vs S2. Acknowledgment of the proteomic data and/or the limitations of transcriptomic data in making conclusions about protein expression levels should be noted.

This information has now been included, we apologize that it was lacking in the original version. However, despite the limitations, we believe that overall the gene expression data fit nicely with our functional measurements. Indeed, older histological studies using antibody stains also reported high lysosomal cathepsin expression in S1 (e.g. PMID: 3091544). Meanwhile, various brush border peptidases were demonstrated to be highly expressed in S2/3 in rodents (e.g. PMID: 7045050; PMID: 6115833), and in humans (PMID: 2866172). Moreover, the appearance of PT brush border peptidases coincides with that of lysosomal enzymes in post-natal kidney development (PMID: 7327946), hinting at an integrated system. These references have been added. What was lacking from all of these studies was an overarching explanation for axial patterns in PT enzyme expression, which we believe our study now provides.

The authors suggest with no evidence in their schematic in Fig 3C that fragment uptake occurs gradually in S2 and is maximal in S3. They cannot image this segment intravitaly, and it is impossible to ascertain from the data in Fig 2I whether there is any uptake in S3 segments. Because S3 segments have high peptide transporter levels but very low expression of Dab2, staining fixed kidneys similar to those in Fig

2 with markers to directly compare tripeptide vs dextran uptake in S2 vs S3 segments would enable the authors to distinguish between fluid phase vs transporter-mediated uptake.

To demonstrate the high S3 uptake of peptides, we have included new overview images in SFig 4, where it can be appreciated that the peptide signal is particularly high in PTs extending into the medulla, but lacking the S2 marker OAT1. This point is also reinforced by the new dextran experiments mentioned above.

The authors suggest that lysine competes for megalin binding, but the role of effect of lysine on protein reabsorption in mice is irreversible and likely complex (see for example Sumpio and Maack, AJP-Renal (1982) 243:F379) Additionally, the time course for imaging in Fig S3b is not noted. If lactoglobulin is fully filtered within 10 min then surely all the lysine must be gone as well? Conversely, if lysine were in fact competing directly for uptake, then the differential conjugation of accessible lysine residues in their lactoglobulin probes may be problematic. The SVD1 kinetics showing differential uptake of low and highly conjugated lactoglobulin in Fig 5i and j would appear to bear this out.

We cannot comment from our study on the mechanism of lysine inhibition and we have rewritten the manuscript to make this clear. However, it has been used in multiple studies to block PT protein uptake in vivo (e.g. PMID: 19261743, PMID: 32123080). We also observed a very clear inhibition of protein reabsorption with lysine. In our experience, blocking PT protein uptake in vivo by other means is quite challenging, due to the very high uptake capacity of S1. The time course of imaging in FS3b has been added – apologies that this was missing. We understand the point the reviewer is making about available lysine residues, but we did not observe major differences in uptake/transition kinetics between moderate/high labeled proteins, suggesting that they were handled similarly.

The authors note that a fraction of degraded peptides processed by the PT is known to be released lumenally, however, a considerable fraction is also released basolaterally. One would predict that this fluorescence would be somewhat concentrated and visible within the interstitium, but this does not seem to be the case.

We did not observe evidence of basolateral release within the time frame of the imaging experiments. However, as we acknowledged in the original discussion, we cannot exclude the possibility that a small amount of undetected basolateral release might occur.

The FIB-SEM images are spectacular and clearly required an enormous effort. A surprising feature is the large apical invaginations, which look very different from those observed in conventional EM images of the proximal tubule. Please provide the fixation and perfusion rate conditions for how the tissue was prepared and comment on whether this might represent an artifact of the tissue preparation or whether this feature is obscured in conventional EM.

We thank the reviewer for their positive feedback. We have provided additional information in the methods concerning the perfusion/fixation. We were also surprised by the size of the apical invaginations, but have no reason to believe that these were an artefact. In single 2D images parts of the invaginations can easily be mistaken for discrete apical vesicles - it was only by reconstruction that we realized they were actually in contact with the lumen, which demonstrates the advantage of FIB-SEM.

How the FIB-SEM data are correlated to the SVD shapes and what these shapes represent in subsequent figures is uninterpretable as described. What is the x axis in Fig 3f and g? Why is low-labeled lactoglobulin visible immediately upon injection whereas high-labeled lactoglobulin visible only after reaching lysosomes? The characterization of these probes in supplemental data shows that the high-labeled version has comparable or higher fluorescence intensity at physiologic pH per molecule.

The x axis depicts normalized distance across the PT cell (from apical to basolateral). This information has now been added, we apologize that it was not clearer before. Regarding the very highly labeled (6x) lactoglobulin species, the in vitro data provided in SFig 2 display relative increases in fluorescence rather than absolute. Nevertheless, the reviewer is correct that some signal can be observed in vivo with this protein before the large increase due to unquenching in lysosomes, and we have adjusted the wording in the manuscript accordingly.

The grandiose conclusion in the abstract that “the kinetics of renal protein processing are remarkably consistent between animals, suggesting a highly conserved mechanism” means only that three (male) mice showed similar results. It seems premature to state based on this that ELS dynamics in vivo “are far less heterogeneous than in non-specialised cell culture models”.

We were truly very surprised at just how reproducible the kinetics of PT ELS function were in living mice, in comparison to the huge variability that is typically seen in cell models, and felt that this was worthy of comment. Although data from only 3 mice are depicted in F5, we imaged many more animals during the course of the study with various different protein injections. Nevertheless, we understand the point that the reviewer is making and have tempered the text accordingly.

The Lowe Syndrome mouse model is described as *Ocr1*^{-/-} but based on the reference provided and the lack of a renal phenotype in *Ocr1* knockouts, this is likely the humanized mouse model that overexpresses human INPP5B in place of the endogenous mouse isoform. The model should be mentioned in the methods sections and potential limitations of this model should be noted.

This information has been added. Our apologies that it was missing from the original manuscript.

The concept that there is axial “remodeling” in *Ocr1* knockout mice implies a functional change in segment identity or in segment length, a finding that is unsupported by the data presented. Rather, their data suggest that a reduction in megalin expression in these mice causes an increase in the axial length across which protein uptake occurs. In fact, this prediction has previously been suggested in a published model that is not curiously not referenced in the study (Glozzi et al. (2020) JASN 31:67).

We understand the point made by the reviewer and have reworded the manuscript accordingly. We have also added the suggested reference, apologies that this was missing before.

Do the authors have data from autofluorescence measurements and morphometry to suggest any change in cortical segment length? The study referenced above quantified shorter S1 segments in zebrafish *Ocr1* knockouts. Is the “remodeling” that the authors propose here congruent with this finding?

Although we did not formally quantify the cortical segment length in KO mice, we did not observe any obvious decrease. The increased protein uptake length we observed is not entirely congruent with the zebrafish model. These discrepancies might be explained by species differences and/or differences in the genetic model. We have discussed this issue in the revised manuscript. While preparing this revision, another study was published on mice lacking the ELS gene Ehd1 (PMID: 35149593). These mice also have a substantial endocytotic defect, and although not formally quantified, it appears that protein uptake length is increased in these animals, suggesting that it might be a generic response.

Data from the Clc5 heterozygote knockout mice are challenging to interpret. The “x-” ROIs marked in Fig 6B appear to be in a region where the tubule is curving away from the plane shown. More details about what was done might clarify this point.

Due to the 3-D morphology of tubules, it is indeed often difficult to capture an entire section within a single focal plane. Analysis was therefore performed on regions that were clearly in plane (denoted by inlay box).

Minor issues:

The title and abstract do not support the conclusions

In this study we have tracked the uptake and processing of proteins in the kidney in real time and mapped events to specific tubular structures. Therefore, we believe the title is appropriate. However, we have rewritten the abstract in the light of some of the comments above.

The authors suggest in the introduction that ligand binding to megalin and cubilin triggers endocytosis, whereas, the general consensus is that trafficking of these receptors is constitutive. If the authors have evidence that this is the case, it should be cited.

This has been rewritten.

There is a disconnect between the references cited in the text and the list at the end.

Apologies, this has been corrected.

S2 segments in Fig 1c,e,h,and j should be indicated. Tubules in Fig 1k appear significantly dilated compared with the same section in 1j. Is this evidence of renal distress at this late time point?

S2 segments have been labeled in Fig 1. We understand the point the reviewer is making concerning the tubules in fig1k, but do not have actual evidence that the tubular lumen was altered, since we did not label the apical membrane. However, from experience we would not regard this as a sign of stress - in unhealthy animals the tubules are typically rather collapsed.

The beginning of the results section is not specified.

This has been corrected.

There are occasional typos in the figures

Our apologies, these have been corrected.

There are some clunky labels- eg, “surface reached by the primary urine” in Fig 4b.

This has been changed to “Apical membrane invaginations”.

What do the boxes in Fig 4a represent? They appear to be largely luminal.

As stated in the legend, the boxes represent single tiles, which were stitched together to provide the overview images.

The suggestion that there are some “potential limitations” to the approaches seems understated at best.

A significant obstacle in this field is the challenge of obtaining quantitative in vivo data, and this team has enviable expertise in this area that enables them to address questions that can be studied by a handful of groups worldwide at best. Although science is ultimately self-correcting, these investigators have a particular obligation to objectively evaluate the strengths and limitations of their studies, specifically because they are so difficult to perform, replicate, and extend. A more balanced treatment of the data presented here, including a tighter focus on the FIB-SEM and subsequent figures, would considerably enhance the value of this work.

We understand the point that the reviewer is making and have expended the discussion on study limitations accordingly. As mentioned above in the response to reviewer 1, we have streamlined the section on FIB-SEM.

Reviewer #3 (Remarks to the Author):

This manuscript from Polesel et al describes the labelling of proteins and peptides with fluorescent dyes in order to image their progression through the proximal tubules of living mice. The authors use Singular Value Decomposition (SVD) analysis to deconvolute spatially and temporally composite signals and identify a sequential activity of distinct proximal tubule sub-segments for protein degradation.

The results are interesting; however, a large amount of data is presented, and the explanation of the data presented in the figures in the main text is very brief. Clarity and readability of the data would be improved in a longer form article, i.e. a research article rather than a communication. The discussion is in general well-reasoned, however there are concerns to be addressed.

We thank the reviewer for their positive feedback. Please see our general comment above, the manuscript has now been expanded to include more explanation.

- The statement in the abstract (also in the main text): “We also find that the kinetics of renal protein processing are remarkably consistent between animals, suggesting a highly conserved mechanism” should be reworded as it can be interpreted that different species of animals have been investigated, which is not the case.

Please see our response to reviewer 2 above, this has been reworded.

- The authors refer to “sophisticated computational analysis” in the abstract and elsewhere, however more specific reference to SVD analysis is preferable.

This has been rewritten.

Regarding the generation and characterisation of labelled proteins and peptides:

- The number of investigated peptides (purified and labelled) does not appear to be stated in the text or supplementary information. The methods state that peptides were individually purified and characterised: “Exact peptide mass and purity (>95%) were confirmed” and “The labelling position was confirmed by ESI-MS/MS analysis.” A summary of these data and the associated spectra and chromatograms should be provided in the supplementary information.

This information has been provided in a new figure (SFig 7).

- The methods also state: “Mobile phase gradients were adapted individually for each peptide.” However, this information is not given. Reproducibility would be improved were this to be included in a summary table of identified peptides.

This information has been added to the methods.

REVIEWERS' COMMENTS

Reviewer #1 (Remarks to the Author):

While I appreciate the additional explanation of the SVD analysis in this revised manuscript, I remain skeptical of the additional information that it provides to the study, and the added value of the electron microscopy.

Nonetheless, per my original review, I am impressed by the ingenious studies providing compelling evidence of sequential protein processing by S1 and S2 segments. In the revised text, I believe that the authors have sufficiently softened their interpretations of the relationship between the SVD base vectors and the structures detected in EM images. With these revisions, the manuscript stands on the strength of the observations of sequential processing in the proximal tubule, allowing the reader to decide for themselves how to interpret the SVD analysis.

Reviewer #2 (Remarks to the Author):

The revised manuscript is significantly improved over the original version. The results are approachably written, methods are more complete, and limitations are more clearly discussed. The new data in figure s4 provide strong support for transport mediated uptake of tripeptides in the later regions of the proximal tubule. I have only a few minor points to suggest:

The authors continue to claim that that fluorescent low molecular weight proteins are fully cleared from the blood in an astonishing 10 min (p.5). The lack of detectable fluorescence in the bloodstream could well be due to the relative dilution/fluorescence sensitivity. I can find no references that suggest such a rapid clearance time for even very small radiolabeled proteins. I don't think it would detract from the significance of the conclusions for the authors to acknowledge the possibility that at least some of the fluorescence in S2 at later times may reflect endocytic uptake of intact protein.

The vacuoles, invaginations, and luminal diameter seen in the ultrastructure images and FIB-SEM are very large compared with some other published studies. It is clear that fixation methods can have a significant effect on PT morphology, and the authors should point this out as a caveat.

The authors suggest that albumin might be handled differently due to binding of Fc receptor. Alternatively, binding to both megalin and cubilin, rather than to cubilin alone, might explain differences in ligand uptake (eg PMID: 32200668, PMID: 35178707). The authors should mention this possibility.

While the authors correctly note that the megalin-expressing PT section in zebrafish is apparently shortened, the same study also described a kinetic model that predicted uptake of albumin and low molecular weight proteins over a longer segment of the PT in humans. The authors should mention the congruence between the prediction and their actual data in the humanized mouse model.

The authors note that the high urinary excretion of small peptides in Lowe syndrome is predicted by their model- can they please elaborate why?

Of note, the data here reveal an unappreciated role for the S3 segment in recovering released peptides. This represents a significant advance in the assignment of any function for this segment, and the authors might want to stress the novelty of this conclusion.

Congratulations on a lovely study.

Ora Weisz

Reviewer #3 (Remarks to the Author):

All my previous concerns have been addressed.

We thank the reviewers for their supportive comments, which are very much appreciated. Responses to further issues raised are provided below. Changes to the manuscript are in red.

Reviewer #1 (Remarks to the Author):

While I appreciate the additional explanation of the SVD analysis in this revised manuscript, I remain skeptical of the additional information that it provides to the study, and the added value of the electron microscopy.

Nonetheless, per my original review, I am impressed by the ingenious studies providing compelling evidence of sequential protein processing by S1 and S2 segments. In the revised text, I believe that the authors have sufficiently softened their interpretations of the relationship between the SVD base vectors and the structures detected in EM images. With these revisions, the manuscript stands on the strength of the observations of sequential processing in the proximal tubule, allowing the reader to decide for themselves how to interpret the SVD analysis.

Reviewer #2 (Remarks to the Author):

The revised manuscript is significantly improved over the original version. The results are approachably written, methods are more complete, and limitations are more clearly discussed. The new data in figure s4 provide strong support for transport mediated uptake of tripeptides in the later regions of the proximal tubule. I have only a few minor points to suggest:

The authors continue to claim that that fluorescent low molecular weight proteins are fully cleared from the blood in an astonishing 10 min (p.5). The lack of detectable fluorescence in the bloodstream could well be due to the relative dilution/fluorescence sensitivity. I can find no references that suggest such a rapid clearance time for even very small radiolabeled proteins. I don't think it would detract from the significance of the conclusions for the authors to acknowledge the possibility that at least some of the fluorescence in S2 at later times may reflect endocytic uptake of intact protein.

We understand the point that the reviewer is making and have acknowledged this point.

The vacuoles, invaginations, and luminal diameter seen in the ultrastructure images and FIB-SEM are very large compared with some other published studies. It is clear that fixation methods can have a significant effect on PT morphology, and the authors should point this out as a caveat.

This has been acknowledged in the revised discussion.

The authors suggest that albumin might be handled differently due to binding of Fc receptor. Alternatively, binding to both megalin and cubilin, rather than to cubilin alone, might explain differences in ligand uptake (eg PMID: 32200668, PMID: 35178707). The authors should mention this possibility.

This has been mentioned.

While the authors correctly note that the megalin-expressing PT section in zebrafish is apparently shortened, the same study also described a kinetic model that predicted uptake of albumin and low molecular weight proteins over a longer segment of the PT in humans. The authors should mention the congruence between the prediction and their actual data in the humanized mouse model.

This has been mentioned.

The authors note that the high urinary excretion of small peptides in Lowe syndrome is predicted by their model- can they please elaborate why?

An extension of the PT length performing whole protein uptake/degradation could lead to an effective shortening of the remaining PT available to perform peptide uptake/degradation, and thus an increase in urinary excretion. This has been clarified in the revised manuscript.

Of note, the data here reveal an unappreciated role for the S3 segment in recovering released peptides. This represents a significant advance in the assignment of any function for this segment, and the authors might want to stress the novelty of this conclusion.

This has been stressed in the revised discussion.

Congratulations on a lovely study.
Ora Weisz

Reviewer #3 (Remarks to the Author):

All my previous concerns have been addressed.